# Mapping the adaptation solution space – Lessons from Jakarta

Mia Wannewitz[1], Matthias Garschagen[1]

[1] Department of Geography, Ludwig-Maximilians University Munich, 80333 Munich, Germany

*Correspondence to*: Mia Wannewitz (mia.wannewitz@lmu.de)

**Abstract.** Coastal cities are under increasing pressure to adapt to climate change. They suffer from the severe effects of increased frequencies and intensities of coastal hazards, particularly flooding, whilst oftentimes continuing to sprawl into hazard exposed areas and grow beyond the pace of sufficient infrastructure development. Even though these problems have been quite well understood for a while, there is still comparatively little knowledge on the scientific assessment of the solution space, i.e. on the options available for adaptation and the ways in which they are being perceived, framed and evaluated in the
scientific literature. Focusing on Jakarta, this study presents findings from a systematic assessment of peer-reviewed scientific literature on the adaptation solution space with regard to current and future flooding. Jakarta is chosen as a case study since it is amongst the cities with the highest flood risk and adaptation pressure globally, whilst also being one of the most heavily researched coastal cities in this regard, certainly in the Global South. Based on a structured key word search, we assess 311 articles. Results indicate that the perceived solution space is skewed towards hard protection against flooding, while measures
to accommodate flooding or retreat from exposed areas are less widely considered in the scientific debate. Soft adaptation measures for the reduction of social vulnerability receive less attention in the literature than those measures targeting the taming of flood hazards, often through engineering solutions. Likewise, hybrid adaptation approaches, which combine soft and hard measures in a complementary way, are only rarely considered. Looking into the future, the findings suggest that despite the importance of hard flood protection as a main adaptation solution in Jakarta, other fields of the solution space
deserve increased scientific attention. This relates in particular to urgently needed feasibility and effectiveness assessments of ecosystems-based solutions for flood mitigation and adaptation options targeting social vulnerability. While the empirical results are specific to Jakarta, heuristic observations from research on other coastal cities suggest that similar scoping exercises of the predominantly perceived solutions space might be of relevance in many cities beyond Jakarta.

## 1.  Introduction

Many coastal cities around the world suffer from chronic flooding, straining their development (Hallegatte et al., 2013). Looking into the future, risks related to flooding in these cities are set to rise sharply (IPCC, 2019a). This rise in risk is driven by climate change effects (e.g. sea level rise and the increasing intensity of heavy precipitation, river flooding and storm surges) but also effects of urbanization itself (e.g. land subsidence, urban sprawl into flood-prone areas or the accumulation of people and infrastructure) (Tellmann et al. 2020; Wolff et al. 2020). As a result, coastal cities are under increasing pressure to
adapt over time and in some instances transform fundamentally (IPCC, 2019a; Revi et al., 2020). While this is not a new phenomenon, even well-researched cities like Jakarta, Indonesia, with sound scientific knowledge on their flooding problems and considerable efforts to improve flood risk management and climate change adaptation, keep suffering from flooding year after year.

Persistent flooding hints towards the existence of "adaptation gaps" (UNEP, 2018). In order to better understand and address
such gaps, this study looks at scientific research not only on past, current and future trends in flood risk but particularly adaptation for the case of Jakarta. Research on this city shows a pattern which appears to be typical for coastal cities: Whilst the problem of flooding and its drivers has been quite well researched for Jakarta (e.g. Abidin et al., 2015; Asdak et al., 2018; Budiyono et al., 2016; Garschagen et al., 2018; Latief et al., 2018; Mishra et al., 2018; Moe et al., 2017; Ward et al., 2011), much less attention has been given to analysing different potential adaptation options. The main focus of this paper is therefore
on examining how the so-called "adaptation solution space" for Jakarta is being covered and framed in the literature. The

concept of "solution space" for adaptation to climate change has been receiving increasing attention since the IPCC's Fifth Assessment Report (WGII AR5) (IPCC, 2014). The adaptation solution space can broadly be understood as being made up of potential adaptation options including their synergies and trade-offs as well as barriers and enablers. Assessing the adaptation solution space, including the feasibility, effectiveness and adequacy of different adaptation options – and their combinations over time – is essential for informing the composition of adaptation pathways. The solution space for climate change adaptation therefore represents a socially constructed, multi-dimensional space of opportunities for adaptation that determines "why, how, when and who adapts to climate change" (Haasnoot et al., 2020:36), restricted by hard and soft limits to adaptation (Dow et al., 2013). This study assumes that the scientific discource is one important arena in which the soluation space is formed and shaped. The question which adaptation solutions are being conceived in science and how they are being scientifically evaluated has great influence on the practical and political debate shaping actual solution decisions (Haasnoot et al., 2020). This paper therefore focuses on the assessment of scientific literature and how adaptation solutions are being perceived and treated in them. Further research might complement this analysis with empirical work on adaptation perceptions amongst, for example, practitioners, policy-makers and affected actors.

Jakarta represents an example worthwhile analysing. The city's flood problems are so pressing that it can serve as an early laboratory for current and future adaptation challenges many other major coastal cities, which are also located in low elevation coastal zones, e.g. Mumbai, Dhaka, Ho Chi Minh City or Lagos, will also have to deal with (IPCC, 2019a). The combination of continuing urbanization and environmental changes result in increasing risks for the urban population today and in the future, turning the city into one of the most at risk coastal cities globally (Hallegatte et al., 2013; Hanson et al., 2011).

To examine which adaptation options are being considered in the scientific literature and how, the study uses a structured literature review to address the following research questions:

- Which flood risk drivers are considered in research on Jakarta?

- To what extent does scientific research consider different hard, soft and hybrid measures for risk management and adaptation and how are these measures being evaluated?

While the paper is focused on Jakarta, it aims at providing lessons for the assessment and understanding of adaptation solution spaces in other coastal cities facing similar risk and adaptation pressure. The paper is structured into seven parts. Section 2 lays out conceptual considerations around flood risk and adaptation adopted in this study before section 3 briefly introduces the flood context of Jakarta. Subsequently, section 4 describes the methods and data of our analysis. Section 5 presents the results starting with general publication patterns and author affiliations. Subsequently, the results chapter describes drivers of flooding mentioned in the literature before it summarizes hard adaptation measures discussed. Then, soft and hybrid adaptation measures mentioned in the literature are presented. Section 6 summarizes and discusses the results, relating them to the solution space for flood adaptation for Jakarta. It also presents identified gaps. Section 7 offers key conclusions and an outlook.

## 2.    Conceptual considerations – flood risk and adaptation options

Climate change adaptation first and foremost means action to limit and reduce climate risks (Garschagen et al., 2019). In order to understand whether and how adaptation is being framed and perceived, it is therefore necessary to concentrate on the links between adaptation and risk, we argue. In other words, to unpack the adaptation solution space – or rather how it has been discussed in the scientific literature in this case – one therefore has to ask whether and how different adaptation options are considered to take effect on the different components and drivers of risk (Garschagen et al., 2019). In addition, a key question is which factors of risk might be underrepresented in the current adaptation literature. For these reasons, understanding how risk is being produced and composed is essential.

In this study, we draw on a few decades of risk research and understand risk in line with current concepts used in the IPCC to be a function of hazard, exposure and vulnerability (Wisner et al., 2004; IPCC, 2012). In that, hazards can be defined as (environmental or climate-related) events or processes with the potential to cause damage and harm (adapted from Weyer et al., 2019 in IPCC, 2019b). In the case of Jakarta this predominantly means floods. Exposure is understood as the presence of

assets or activities (social or environmental) in the spatial, temporal and/or functional reach of hazards. Exposure therefore has a hybrid character as it can be altered by environmental changes (e.g. sea level rise) as well as the socio-economic change (e.g. urban sprawl). Vulnerability refers to the propensity or predisposition to be adversely affected if exposed to a hazard (adapted from Weyer et al., 2019 in IPCC, 2019b). For example, the degree of susceptibility of livelihoods or infrastructure to suffer harm from floods contributes to vulnerability.

Adaptation links into the causal fabric of risk by aiming to reduce existing as well as future vulnerability (e.g. through health care or other social security programmes), exposure (e.g. through planned retreat from hazard-exposed areas), and/or, where possible, hazards (e.g. by limiting flood intensity through retention areas) (Garschagen et al., 2019). Overall, adaptation can hence be defined as the process of adjustment to actual or expected climate change and its impacts, in order to moderate harm or, where possible, even exploit beneficial opportunities (adapted from Weyer et al., 2019 in IPCC, 2019b). Addressing the

different risk components (hazards, exposure and vulnerability) involves assessing and selecting options for policy and action (Garschagen et al. 2019). Such decision-making entails evaluation of the effectiveness, efficiency, efficacy and acceptance of actions (ibid.). Limits to adaptation apply where available options do no longer allow actors to secure valued objectives, functions or assets from intolerable risk (Dow et al., 2018). While coastal cities might reach technical limits of adaptation only rather late (e.g. in terms of the engineering limits theoretically applying to coastal protection), financial, social and institutional

barriers and limits are expected to be reached much earlier (Oppenheimer et al., 2019).

    Risk analysis is often split into understanding the hazard and understanding other driving forces – including physical and/or social exposure as well as vulnerability. For a long time, scientific research has focused primarily on understanding the hazard with the objective to control nature and protect people from it (Hewitt, 1983; Wisner et al., 2004; IPCC, 2012). In the same vein, the role of "hard" adaptation solutions, which aim at controlling flood hazards and protecting exposed elements, have

grown in importance, becoming a centrepiece of the predominant paradigm for risk reduction in the second half of the 20[th] century (Hewitt 1983; Wisner et al., 2004). Hard adaptation measures in the context of flood risk reduction are mostly large-scale engineered human-built structures, e.g. floodwalls or storm barriers (Sovacool, 2011; Oppenheimer et al. 2019; Du et al. 2020). While they often meet their objective to protect people and systems from harmful events and are widely considered an important element within portfolios of risk reduction measures especially in coastal cities (Oppenheimer et al., 2019), they do

rarely work towards reducing underlying hazard drivers or social vulnerability and they often entail significant downsides. First, they tend to be technologically complex – often prone to failure – and very cost-intensive (Sovacool, 2011). Second, they are comparably inflexible as concrete structures remain for a long time. This can be challenging in the face of high levels of uncertainty regarding climate change and dynamic trends in its impacts, which mean that they need constant assessment and sometimes costly updates (David et al., 2016). Third, hard measures on aggregate generate comparatively little co-benefits

and, depending on the planning and implementation process, have even been harmful to local communities and ecosystems (ibid.; Sovacool, 2011). And lastly, infrastructural measures might give people a false sense of security, increasing the overall damage potential in case of failure (IPCC, 2012). Risk reduction and adaptation regimes centred around hard protection predominantly or exclusively have therefore been problematized for their emphasis on technocratic fixes for solving symptoms rather than causes of risk (Ribot, 2011; Garschagen, 2014; Solecki et al., 2017); paving the way towards addressing the need

for changing the protection paradigm towards more holistic risk management approaches (e.g. Viero et al, 2019).

    Next to hard adaptation, the importance also of soft adaptation measures has therefore been emphasized since a while especially for reducing socio-economic vulnerability (Wisner et al., 2004; Ribot, 2011; Solecki et al., 2015) or for absorbing residual risk remaining beyond hard measures (Du et al., 2020). In contrast to hard adaptation, soft adaptation includes an emphasis on ecological and institutional responses, notably ecosystems-based approaches and institutional adjustments e.g. in terms of land-

use planning, building codes, social protection or awareness raising. Soft adaptation is less clearly defined as hard protection, meaning that also the consideration of properties, advantages and disadvantages of measures belonging to soft protection is multifaceted. Yet, a number of overall observations have been suggested in the literature, notably that soft protection is focused on empowering and capacitating local communities to respond to changing hazards and is often based on modular technologies which do not require large outlays of capital or human resources (Sovacool 2011). Yet, this is not to argue that certain soft

measures also require a large amount of central planning, investment and steering, e.g. in the case of large-scale wetland or mangrove restoration.

Hard and soft adaptation measures are often combined and can both be mapped onto the main response types against sea level rise and coastal flooding as used by the IPCC, i.e. protection, accommodation, advance, retreat (Oppenheimer et al., 2019). However, certain clusters can be observed, e.g. in that protection typically relies on hard measures whereas accommodation typically also requires a stronger integration of soft measures. Assessing coastal adaptation approaches and potentials across the globe, the IPCC stresses that hard adaptation is technologically feasible and economically efficient for coastal cities and therefore will continue to play a central role in adapting such cities further (Oppenheimer et al., 2019). However, the IPCC also stressed that hard protection does not come without disadvantages and raises questions of affordability, particularly in poorer regions of the world (ibid.). There is therefore high agreement that hybrid approaches, combining different hard and soft approaches, is a promising way forward in many coastal settings (ibid.). For Shanghai, for instance, hybrid approaches of combining hard storm-surge barriers with wetland development and wet-proofing of infrastructure have been assessed to bring about the highest potential for overall risk reduction (Du et al., 2020). An example from Padova (Italy) shows how proper floodgate operations are ensured by including the end-user in designing and implementing control structures and protocols (Mel et al., 2020).

The above considerations mean that measures of both types jointly make up the so-called solution space for climate change adaptation which can be understood as a flexible space spanning across multiple dimensions (biological, political, institutional, socio-economic, cultural), scales and actors, containing all potential solutions for climatic risks (Haasnoot et al., 2020). The solution space is confined by soft and hard limits (Dow et al., 2013) and can hence change in form depending on internal and external influencing factors (Haasnoot et al., 2020). This study assumes that scientific research in the field of flood risk management and adaptation represents one of these influencing factors. Through scientific inquiry, scholars assess and evaluate potential flood adaptation options from many different perspectives, creating a diverse and constantly widening landscape of adaptation options, which are readily available for consideration of decision-makers. Accordingly, scientific perspectives play a vital role in shaping the actual solution space.

## 3. Brief overview of Jakarta's flood risk, its root causes, impacts and recent management

Urban flooding has a long history in Jakarta. The city is naturally prone to coastal hydrological hazards due to its geographical location in a low-lying coastal area facing the Java Sea, and with more than 13 rivers, including the Ciliwung river, flowing through it (Marfai, 2015). Urban flooding in Jakarta is most severe when heavy precipitation, high run-off rates and storm and/or high tide levels coincide (Garschagen and Surtiari, 2018). In the future, climate change is expected to increase Jakarta's "natural" drivers of flood risk, mostly through sea level rise and the increasing potential for heavy precipitation in the entire Ciliwung catchment area (Mishra et al., 2018; Januriyadi et al, 2018).

However, besides natural drivers of flooding, there are also multiple human-made causes which significantly contribute to the city's flood problem. First, continuous population growth, urbanization and land use changes in Jakarta as well as in its surrounding areas, including the upstream area of the Ciliwung River, have significantly altered the hydrological system and run-off patterns (Garschagen et al., 2018). Urban densification processes led to a degradation of the urban drainage system as river and canals were filled and floodplains paved over, reducing the retention and discharge potential (ibid.). Second, the narrowing of urban water ways due to informal settlements along the banks of the rivers, sedimentation and pollution with waste have further reduced water flow capacities and urban drainage (e.g. Mathewson, 2018). Third, land subsidence is a key driver of the city's flood problem because it exacerbates the impacts of precipitation and sea level rise (Salim et al., 2019). Today, 40% of Jakarta's urban area lies below sea level (Marfai, 2015; Salim et al., 2019). Subsidence has four major drivers: excessive groundwater extraction, natural consolidation of soil, increasing infrastructure and building load, and tectonic activities (Abidin et al., 2015).The impacts of floods, resulting from the interplay of both, natural and human-made drivers as described above, represent a strain on the city's development until today. Major flood events hit the city in the 21[st] century; i.e. in 2002, 2007, 2013, 2014 and 2020. All of them resulted in several deaths, up to 500.000 evacuees as well as massive direct and indirect economic losses related to infrastructure damages, reduction of productivity as well as business value losses (e.g. Budiyono et al., 2015; Djalante et al., 2017; Octivianti and Charles, 2019). The 2007 floods stand out in extent and severity. The floods submerged more than 60% of the city, causing an unprecedented flood extent, fatalities, damages and

losses (Texier, 2008). They resulted from the confluence of high precipitation in the city, water run-off from upland areas and a strong spring tide pushing water into the city from the sea (Octavianti and Charles, 2019; Garschagen et al., 2018). This extreme event, driven by compound flood drivers including the seaward intrusion of water, can be described as a demarcation point, triggering a paradigm shift in flood risk management in so far that sea level rise was from then on portrayed to be one of the risk drivers (Garschagen et al., 2018; Octivianti and Charles, 2019). While prior to the 2007 event the city government mostly focused on protecting the city from inland flooding, it then developed the so-called Jakarta Coastal Defence Strategy (JCDS) to also protect the city from coastal hazards (Garschagen et al., 2018). Since then, the city government concentrates its flood risk management on four key infrastructural measures; namely river and water way regulation, including dredging and clearance of river banks, canalization, expansion of water reservoirs and the development of a massive sea wall including land reclamation (ibid., Octavianti and Charles, 2019). The so-called Great Garuda is one of multiple flood protection measures of the National Capital Integrated Coastal Development (NCICD) plan adopted in 2014. Its shape resembles a Garuda, the Indonesian national bird (cf. Figure 1). Developed by a consortium of Dutch and Indonesian planning consultancies, the "giant sea wall" is supposed to protect Jakarta's bay area from the sea. In combination with other infrastructural measures of the NCICD masterplan, the bay area will be completely re-invented and developed as a new residential and business district (Garschagen, Surtiari, et al., 2018), aiming at marrying coastal protection with urban development while at the same time addressing land subsidence – a problem that started receiving increased attention particularly after major floods in 2013 and 2014. The question to which extent the entrie project with all its elements will be implemented is subject of continued debate (cf. Garschagen et al. 2018; Minkman et al. 2021).

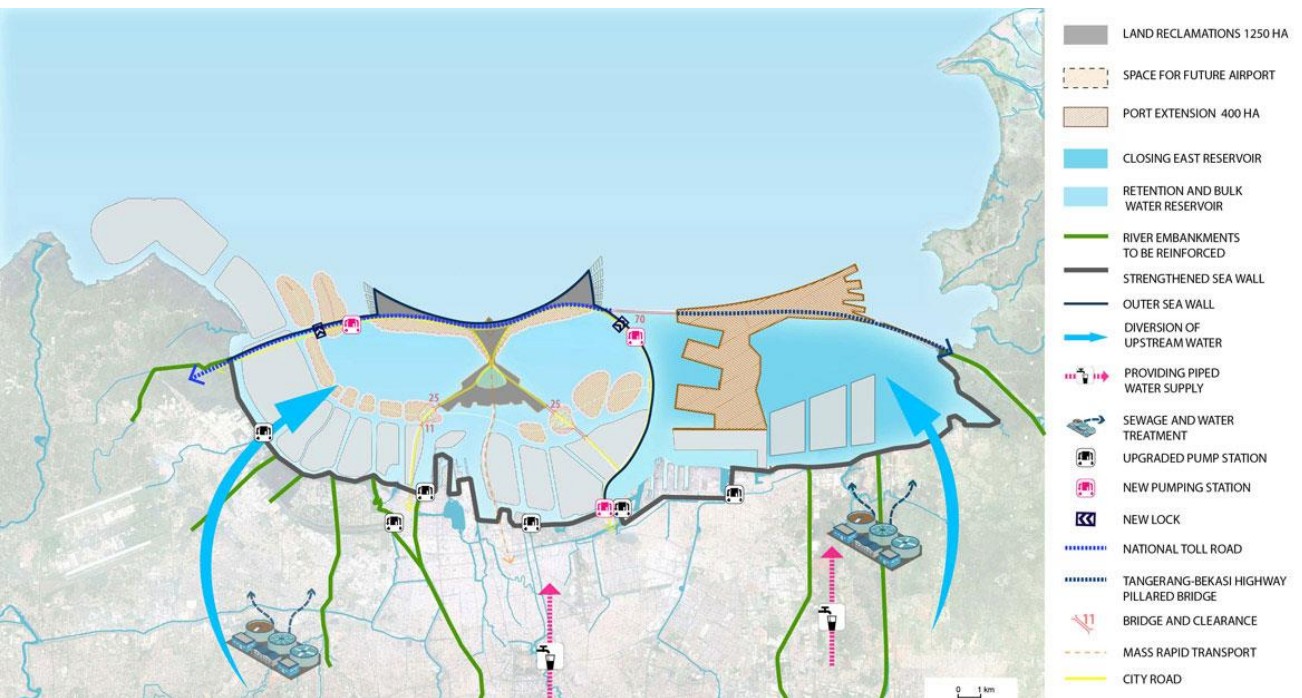

*Figure 1: The Great Garuda project in its originally planned version (taken from Mezzi, 2016, based on the NCICD Master Plan 2014 Coordinating Ministry for Economic Affairs of the Republic of Indonesia, 2014)*

## 4. Methods and data

This study builds on a two-tier analysis scheme. First, a systematic search for English scientific literature in the citation database Scopus was conducted, aiming at getting an overview of the state of research on adaptation and flood risk reduction in Jakarta. While capturing the majority of the international scientific literature, this selection does not capture studies published in other languages such as Indonesian. Nonetheless, the scoping of articles as well as the discussion of our approach and preliminary findings with Indonesian colleagues suggest that our approach is able to capture the ongoing scientific debate quite comprehensively. This is to large part due to the fact that Indonesian scholars today quite actively contribute to the English-language peer-reviewed scientific literature, thereby transporting research results of many local to regional studies into that body of knowledge (Djalante, 2018; see also section 5).

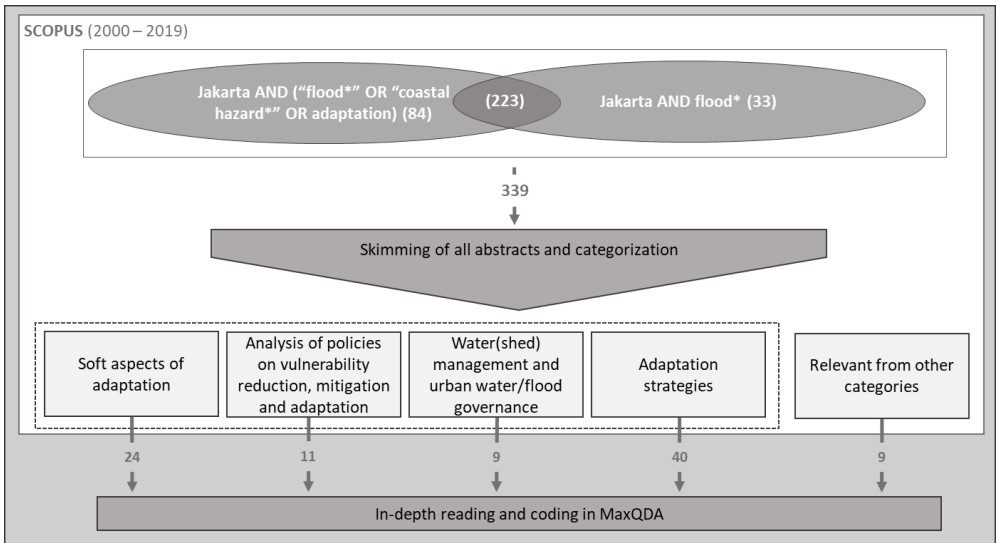

*Figure 2: Literature search, extracted categories and numbers of resulting publications*

As shown in Figure 2, the Scopus literature database was searched with a deliberately broad combination of the search terms Jakarta AND ("flood*" OR "coastal hazard*" OR "adaptation"), limited to roughly the past 20 years (2000 - 2019). In total, 339 publications in English language (see Appendix A) resulted from the structural searches after excluding non-relevant research fields (medicine, mathematics, chemistry, biochemistry, nursing, chemical engineering, dentistry, immunology, veterinary); building the basis for the following filtering and analyses. Seven papers could either not be accesses or no full-text versions could be found (Appendix B). In a first step, all abstracts of resulting publications were skimmed and topical categories were developed, resulting in an inductively derived thematic scheme (see Table 1 and Appendix C). Based on the number of papers per category as well as their content, we obtained an overview of Jakarta's flood risk and adaptation research landscape and its centres of gravity. It is taken as an indication for whether and how different fields of scientific inquiry contribute to the assessment of flood risk and adaptation solutions. Of course, the number of papers alone does not sufficiently evaluate the importance of a study field for framing the solution space and shaping political decision-making. Nonetheless, we argue that the number of publications can be one proxy – amongst others – for evaluating the intensity of a certain debate, in this case the engagement with particular flood drivers and adaptation solutions.

What is more, our analysis does not only draw on a bibliometric assessment but also on the content analysis of a subset of papers. In line with the research interest of our paper, we selected for a detailed content-analysis 93 publications which deal with otherwise underrepresented segments of the adaptation solution space, i.e. those focusing on soft and mixed adaptation options, including soft factors of adaptation and a focus on social vulnerability, policy analysis and integrated management (categories 1, 2, 7 and 8 of Table 1). The analysis was guided by the research questions (section 1). It follows the principles of

a qualitative content analysis (Mayring and Früh, 2002) and draws on a combination of deductive and inductive codes (see Appendix C and D).

*Table 1: Categories of publications*

| Code | Topic | Description |
|---|---|---|
| 1 | **Soft factors of adaptation** | Papers on soft factors that influence adaptation such as psychology, behavior, culture, understanding of risk, how risk is framed, willingness to pay for ecosystem services, participation in planning. |
| 2 | **Policy and legal analysis** | Papers focusing on e.g. institutional analysis, national policy, legal frameworks of risk management, vulnerability reduction and adaptation |
| 3 | **Hard adaptation** | Papers that exclusively look at hard physical adaptation measures such as the Great Garuda Project, infrastructure for rainwater harvesting, polder, dikes and flood barriers, embankments and river diversions. |
| 4 | **Flood models & flood mapping** | Papers that present quantitative precipitation models, subsidence models, flood loss estimation models, urban drainage models, flood cost analysis, urban expansion and its effects, sea level rise models, community-based flood risk mapping, shoreline retreat models. |
| 5 | **Land-use (change) impact on flooding** | Papers that examine the criticality of watersheds or land-use change and its impacts on flooding with the help of quantitative models or qualitative case studies. |
| 6 | **New data types** | Papers that investigate the potential of using new data sources like social media, big data or high resolution data or new data generating formats such as crowd-sourcing or e-participation for flood risk mapping and analysis. |
| 7 | **Watershed management and water governance** | Qualitative analyses of reasons for flooding, water pollution, incl. drinking water source analysis/model. |
| 8 | **Soft and hybrid adaptation** | Papers that focus on soft and hybrid adaptation strategies incl. soft measures, local/community-led adaptation, firms and adaptation, resettlement/relocation, alternative energy sources, disaster management and urban adaptation planning. |
| 9 | **Early warning** | Papers that present GIS-based early warning systems, risk communication and information needs during disasters. |
| 10 | **Decision support systems** | Decision support system for location of warehouses, disaster information management system, socio economic vulnerability index, hydrological infrastructure flood vulnerability index, integrated assessment framework for subsidence. |
| 11 | **Qualitative risk descriptions** | Papers presenting information on flood events and impacts or evolution and impacts of land subsidence. In contrast to category 4, no quantitative hazard or risk models are employed. |

**5.   Results**

As depicted in Figure 3, scientific research on flooding in Jakarta has been rapidly rising since the year 2015. Between the beginning of 2015 and today, almost 3 times as many documents were published compared to the period of 2000 to 2014. In

comparison to global flood risk research, which increased steadily over the years, this represents a remarkable spike, hinting towards the relevance for the topic of flooding in Jakarta as well as an increasing scientific interest.

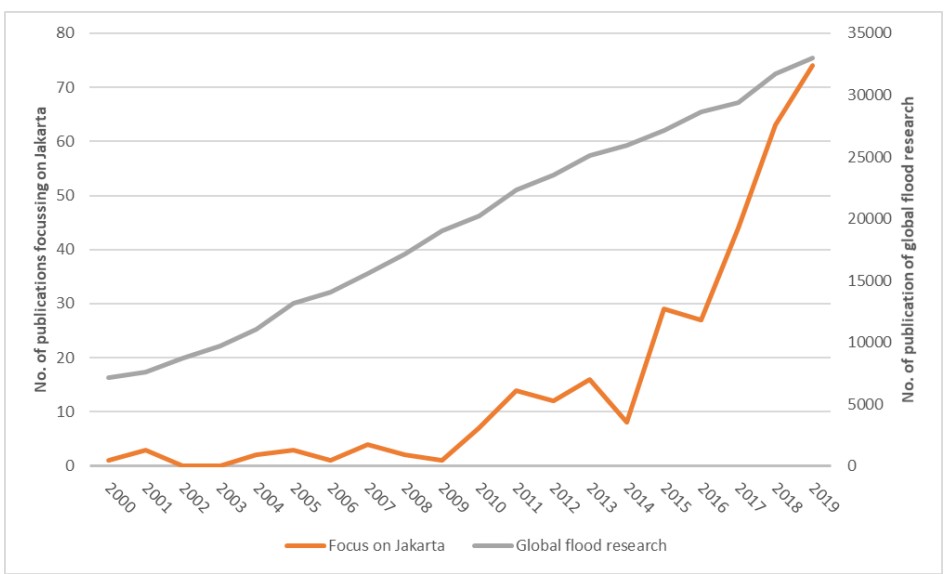

*Figure 3: Number of scientific publications in flood and adaptation research in Jakarta and globally[1] (2000-2019) in Scopus*

Research attention was especially high among Indonesian scholars. As visualized in Table 2 almost 50% of first authors are affiliated with Indonesian institutions. With regard to international attention to Jakarta's flooding issue, most first authors are

245 affiliated with Japanese and Dutch institutions, each making up roughly 11% of the resulting publications, followed by researchers from Germany, the US, Australia, and Singapore.

*Table 2: Number of publications sorted by the location of affiliation of the first author*

| Country of affiliation | No. of publications | % |
|---|---|---|
| Indonesia | 218 | 48.6 |
| Japan | 48 | 10.7 |
| the Netherlands | 47 | 10.5 |
| unknown | 26 | 5.8 |
| Germany | 20 | 4.5 |
| USA | 18 | 4.0 |
| Australia | 17 | 3.8 |
| Singapore | 15 | 3.3 |
| Switzerland | 14 | 3.1 |
| UK | 9 | 2.0 |

---

[1] Searches in Scopus: 1. Jakarta AND (flood* OR "coast* hazard" OR adaptation"), 2. Flood* OR "coast* hazard" OR adaptation; both for the years 2000-2019 and excluding the following subject areas: medicine, mathematics, chemistry, biochemistry, nursing, chemical engineering, dentistry, immunology, veterinary

| China | 3 | 0.7 |
|---|---|---|
| South Korea | 2 | 0.4 |
| Thailand | 2 | 0.4 |
| Austria | 1 | 0.2 |
| Brazil | 1 | 0.2 |
| Canada | 1 | 0.2 |
| Denmark | 1 | 0.2 |
| EU | 1 | 0.2 |
| France | 1 | 0.2 |
| Greece | 1 | 0.2 |
| Italy | 1 | 0.2 |
| Philippines | 1 | 0.2 |
| Spain | 1 | 0.2 |

According to the subject fields provided by the Scopus citation databank (multiple possible per paper), more than three quarters of all publications include a natural science or engineering perspective (environmental science, earth and planetary sciences, engineering, computer science, energy, physics and astronomy, agricultural and biological sciences, materials science, mathematics, psychology, decision sciences) (cf. Figure 4). Only 15% can be attributed fully or partially to social sciences, and very few include an economics perspective (Business, Management and Accounting, Economics, Econometrics and

Finance). Only two percent are labelled to include an arts and humanities angle.

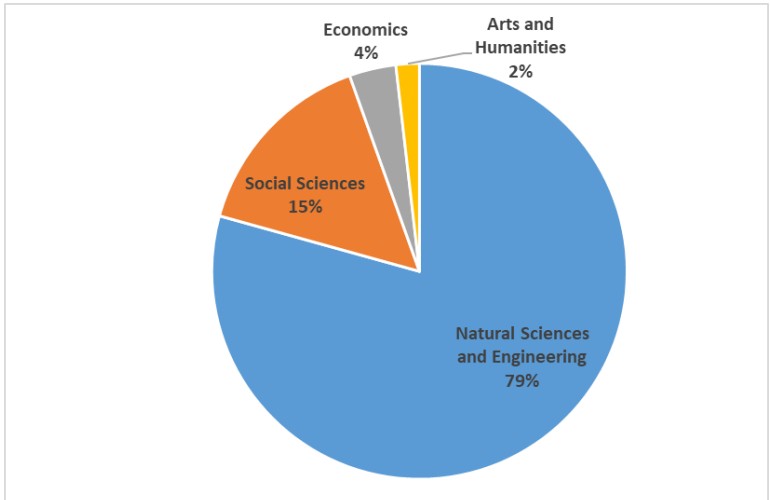

*Figure 4: Subject fields of analyzed publications according to Scopus classification*

To get a clearer and more detailed overview, Figure 5 shows a classification of the resulting publications in terms of the studies' content and focus, building on an inductively developed categorization scheme (Table 1). Some clear clusters and patterns emerge: First, studies on quantitative flood modelling, land use (change) impacts on flooding and hard adaptation options

together dominate the research field, representing almost 50% of all publications (grey fields). This corresponds with the high number of papers in subject fields from the natural sciences (see Figure 4). Second, with around one third of all publications, studies in the areas of soft and hybrid adaptation analyses, soft factors of adaptation, policy and legal analyses as well as watershed management (green fields), together represent another stream of scientific research. The rest of the studies of this analysis are very divers spanning from qualitative risk descriptions, over the employment of new data types for risk analysis and response to early warning systems and other decision support systems as well as land use change analysis in the context of flooding. Together, they represent roughly one quarter of all analysed publications.

*Figure 5: Thematic clusters of publications resulting from the structured literature search*

## 5.1. Understanding the drivers of the flood hazard

As shown in Figure 5, there is a strong focus on flood modelling and mapping, representing almost one third of all papers published since the year 2000. Flood models (Bahtiar et al., 2018; Farid et al., 2012; Formánek et al., 2013;(Jati and Santoso, 2019.; Hurford et al., 2010; Kadri, 2008; Lin et al., 2016; Mishra et al., 2018; Ogie et al., 2016a; Remondi et al., 2016; Rojali et al., 2017; Takagi et al., 2016b; Tambunan, 2018) and non-model-based flood analyses (Asmadin et al., 2018; Priambodo et al., 2018; Syafalni et al., 2015; Yayuk Supomo et al., 2018), rainfall and/or run-off (Aditya et al., 2017; Anggraheni et al., 2018; Anindita et al., 2016; Farid et al., 2011; Hermawan et al., 2017; Kurniawan, 2019; Moe et al., 2017; Otsuka et al., 2017; Rafiei Emam et al., 2016; Riyando Moe et al., 2017) and non-modelled rainfall/run-off analyses (Liu et al., 2015; Nuryanto et al., 2017; Wu et al., 2013) as well as models and analyses of land subsidence (Agustan et al., 2013; Andreas et al., 2019;

Andreas et al., 2018; Chaussard et al., 2013; Koudogbo et al., 2012; Park et al., 2016) all aim at better understanding and simulating the physical factors that cause or influence flooding issues and measuring its impacts in Jakarta. The same holds true for flood damage or estimated losses models (Budiyono et al., 2015, 2016; Fajar Januriyadi et al., 2018; Kurniyaningrum et al., 2019; Marko et al., 2019; Wahab and Tiong, 2017; Ward et al., 2011b; Wijayanti et al., 2017).

With the objective to identify spatial patterns of rainfall (Latifah and Setiawan, 2014), subsidence (Abidin, 2005; Abidin et al., 2015; Prasetyo et al., 2018) and flooding/inundation (Andreas et al., 2017; Latief et al., 2018; Margatama et al., 2018; Nuswantoro et al., 2016; Tambunan, 2017; Ward et al., 2013c) the phenomena were mapped for specific rainfall and/or flood events. Soemabrata (2018) adopts a more comprehensive perspective, developing a flood risk map that also considers vulnerability and urban growth and Padawangi et al. (2016) highlight the role of community risk perception and local knowledge by referring to the use of community-based and participatory flood mapping.

Apart from the latter two exceptions, studies from the categories flood models & flood mapping as well as hard and physical adaptation tend to focus on climatic, hydrological and physical factors contributing to flooding, thereby providing key information on flood drivers and patterns. Such studies are much higher in number than risk assessments that also include non-hydrological risk drivers (see below).

Publications which focus more on soft and hybrid adaptation as well as on social vulnerability issues (i.e. the green categories in Fig. 5) predominantly argue that the flooding problem is not only caused by the local topography, geology, tidal influence and regional climatic patterns that successively change in the course of climate change but also – potentially even more so – by anthropogenic factors (Aerts et al., 2013; Akmalah and Grigg, 2011; Asdak et al., 2018; Batubara et al., 2018; Costa et al., 2016; Esteban et al., 2017; Firman et al., 2011; Garschagen et al., 2018; Garschagen and Surtiari, 2018; Hellman, 2015; Ichwatus Sholihah and Shaojun, 2018; Kadri, 2008; Leitner and Sheppard, 2017; Marfai et al., 2015; Mathewson, 2018; Neise and Revilla Diez, 2018; Neise and Revilla Diez, 2019; Neolaka, 2012; Noviandi et al., 2017; Nurhidayah and McIlgorm, 2019; Octavianti and Charles, 2018, 2019; Padawangi and Douglass, 2015; (Rahayu et al., 2020); Salim et al., 2019; Shatkin, 2019; Sheppard, 2019; Simanjuntak et al., 2012; Simarmata, 2018; Texier, 2008; Varrani and Nones, 2018; van Voorst, 2014, 2016; van Voorst and Hellman, 2015; Ward et al., 2011a, 2013a; Wicaksono and Herdiansyah, 2019; Yoga Putra et al., 2019a; Yuliadi et al., 2016). Three of the most important and frequently mentioned anthropogenic factors are accelerating land subsidence (e.g. Andreas et al., 2019; Colven, 2017; Costa et al., 2016; Fitrinitia et al., 2018; Garschagen et al., 2018; Goh, 2019; Padawangi and Douglass, 2015; Salim et al., 2019; Sari et al., 2018; Shatkin, 2019; Ward et al., 2011b), river clogging due to waste disposal (e.g. Akmalah and Grigg, 2011; Garschagen et al., 2018; Garschagen and Surtiari, 2018; Kadri, 2008; Marfai et al., 2015; Mathewson, 2018; Padawangi and Douglass, 2015; Shatkin, 2019; Simarmata, 2018; Texier, 2008; Varrani and Nones, 2018; van Voorst and Hellman, 2015; Ward et al., 2011b) and land conversions (e.g. Asdak et al., 2018; Batubara et al., 2018; Garschagen et al., 2018; Garschagen and Surtiari, 2018; Kadri, 2011; Marfai et al., 2015; Padawangi and Douglass, 2015; Shatkin, 2019; Varrani and Nones, 2018; Ward et al., 2011b, 2013a).

## 5.2. Hard flood protection measures and their evaluation

Considering the focus on natural or geo-physical drivers of flooding (see section 5.1), it is not surprising that there are many publications that concentrate exclusively on so called structural, engineered or hard flood protection measures. Such measures started being implemented already during colonial times and remain, until today, to be a main pillar of Jakarta's approach to mitigate flooding (Colven, 2017; Garschagen et al., 2018; Goh, 2019; Mathewson, 2018; Octavianti and Charles, 2018; Owrangi et al., 2015; Padawangi and Douglass, 2015; Simanjuntak et al., 2012; Ward et al., 2013a). Publications from the category of hard adaptation form a major part of flood risk management research in Jakarta overall (Figure 5). They focus on specific solutions such as levees, dams, dikes and embankments and analyse or model their protective capacity or suitability for flood protection as well as their vulnerability (Mardjono et al., 2018; Mardjono and Setiawan, 2018; Ogie et al., 2016b; Su et al., 2018; Sujono, 2012; Susilo et al., 2019; Suprayogi et al., 2018; Takagi et al., 2016a, 2017; Wurjanto, 2018). Some are concerned with water channelling, retention ponds and drainage systems as a means to mitigate flooding (Indrawati et al.,

2018; Kadri, 2011; Kartolo and Kusumawati, 2017; Mahanani and Chotib, 2018; Mohajit, 2015; Nugroho et al., 2018; Sholichin et al., 2019; Wihaji et al., 2018).

Some authors exclusively focus on the Great Garuda Project the central element of the NCICD masterplan. Besides outlining
the plan and its objectives, a number of studies question its effectiveness regarding flood protection (Badriana et al., 2017) and its potential impacts on the local environment (Rusdiansyah et al., 2018; van der Wulp et al., 2016). Modelled scenarios of flooding with and without the Great Garuda by (Yahya Surya et al., 2019) show that the protection wall would slightly increase wave amplitudes, so that the authors conclude that the project requires improvements to meet its aim of flood protection. (David et al., 2016) point towards the option to complement the seawall project with ecosystem-based adaptation measures.

Studies outside the category of hard adaptation (Fig. 5) mostly adopt a rather critical perspective on infrastructural solutions in general and on the Great Garuda project in particular. The latter is criticized to be a politically and economically driven, technocratic mega project that fails to comprehensively address the flood problem (e.g. Colven, 2017; Octivianti and Charles, 20; Salim et al., 2019). The respective studies do not argue that the project cannot provide any protection from flooding but they are concerned that it neither addresses land subsidence nor other socio-economic factors contributing to flooding; it
therefore does not present a comprehensive and sustainable solution for the flooding problem (e.g. Colven, 2017; Garschagen et al., 2018; Garschagen and Surtiari, 2018; Octavianti and Charles, 2019; Salim et al., 2019; Shatkin, 2019; Wade, 2019). Infrastructural solutions in general are often portrayed as "technocratic fixes", which do not sufficiently address the hazard's root causes, which are argued to stem from socio-economic and structural context conditions and vulnerabilities, so that the problem persists (Colven, 2017; Padawangi and Douglass, 2015; Wade, 2019).


### 5.3. Soft and hybrid approaches to flood risk management and their evaluation

With a less model-driven and engineering-based perspective, another stream of literature is dedicated to describe, analyze and/or propose adaptation strategies and flood governance approaches from a more integrated perspective, i.e. considering also soft measures or hybrid approaches combining soft and hard measures, which, according to the literature, are implemented
through both, state-led and community driven initiatives. The analysed publications provide multiple examples of state-led soft measures. (Amri et al., 2017; Dwirahmadi et al., 2013; Faedlulloh et al., 2019; Hellman, 2015; Sugar et al., 2013; Yoga Putra et al., 2019a, 2019b) for instance mention the government's involvement in community empowerment and capacity building to facilitate and improve climate change adaptation. Other studies point to the involvement of government at different levels in the dissemination of information about flood risk and adaptation options, which help to raise awareness as well as
prepare for and mitigate flooding (Dwirahmadi et al., 2013; Guinness, 2019; Texier, 2008; Ward et al., 2013a). Furthermore, the government's approach of combining hard flood protection infrastructure with relocation of exposed population as in the case of e.g. Great Garuda could be described as a hybrid approach, which is subject of discussion in many publications of this stream of literature (e.g. Colven, 2017; Rusdiansyah et al, 2018; Salim et al., 2019; Wade, 2019).

Local, community- or NGO-led adaptation initiatives are argued to build on a wealth of context-specific knowledge about and
experience with flooding and often compose of soft as well as hard measures (e.g. Bott et al., 2019; Fitrinitia et al., 2018; Marfai et al., 2015; Padawangi and Douglass, 2015; Purba et al., 2018; Simarmata, 2018; Sugar et al., 2013; van Voorst and Hellman, 2015; Yoga Putra et al., 2019b). The review of community-led adaptation efforts revealed a strong focus on the importance of what can be summarized as social capital for adaptation. A number of studies describe the key role of social networks, which allow for sharing knowledge, experience and best practices (Sugar et al., 2013; Yoga Putra et al., 2019a),
which facilitate cooperation and coordination within and among communities, with NGOs and with universities (Fitrinitia et al., 2018; Goh, 2019; Hellman, 2015; Mathewson, 2018; Padawangi and Douglass, 2015; van Voorst, 2014; Yoga Putra et al., 2019a) and which foster mutual support as well as "practices of communing" (Leitner and Sheppard, 2017) such as pooling of resources (Guinness, 2019; Leitner and Sheppard, 2017; Padawangi and Douglass, 2015; van Voorst and Hellman, 2015). Social cohesion within networks has also been directly linked to collective action for adaptation (Rahmayati et al., 2017).
Empirical examples put forward are for instance collective community works - in Indonesia known as "gotong royong" - for e.g. collective action in river monitoring and the issuance of flood warnings (Bahri and W Purwantiasning, 2019; Dwirahmadi et al., 2013; Fitrinitia et al., 2018; Hellman, 2015; Padawangi and Douglass, 2015; van Voorst, 2014, 2016), flood risk mapping

(Dwirahmadi et al., 2013) and the establishment of local, community-based institutions to collectively develop and administer saving schemes and funds used for flood response and recovery (Dwirahmadi et al., 2013; Marfai et al., 2015; van Voorst, 375 2014).

Looking at how soft and hybrid measures are being evaluated in the literature, this review finds that many studies exhibit a rather critical perspective on state-led soft adaptation measures. (Fitrinitia et al., 2018; Mathewson, 2018; Ward et al., 2011a) elaborate that increased state investments into non-structural, soft measures, as described above, are not materializing on the ground. (Hellman et al., 2018a) refer to this as an "implementation deficit", which they find being facilitated by lacking 380 reinforcement of laws and regulations. The state's hybrid approach, i.e. the combination of protective infrastructure and relocation of exposed population, is criticized to not only incomprehensively address flood risk (e.g. Colven, 2017; Garschagen et al., 2018; Garschagen and Surtiari, 2018; Octavianti and Charles, 2019; Salim et al., 2019; Shatkin, 2019; Wade, 2019) but also to cause serious negative effects on the environment and local communities (e.g. Dovey and Achmadi., 2019; Garschagen et al., 2018; Garschagen and Surtiari, 2018; Leitner and Sheppard, 2017; Neolaka, 2012; Rahmayati et al., 2017; Surtiari et al., 385 2017; Texier, 2008; van Voorst, 2016; van Voorst and Hellman, 2015; Ward et al., 2013a) due to the major relocations of informal settlers and urban poor in highly exposed areas at banks of the river and coastal areas in the name of flood mitigation (e.g. Goh, 2019; van Voorst and Hellman, 2015). This state-led approach is described to have accelerated since 2009 (Dovey and Achmadi., 2019; Ichwatus Sholihah and Shaojun, 2018) and some authors even claim these evictions to be a bigger threat to those evicted than flooding itself (Dovery et al., 2019; Hellman, 2015; Saridewi and Fauzi, 2019).

At the same time, studies highlight the potential of soft adaption measures to create co-benefits with other development objectives. Many studies analyse how the adaptation measures' impacts go far beyond mitigating flood vulnerability as they function as social and financial security, addressing a wide range of vulnerabilities (Hellman et al., 2018a; Padawangi and Douglass, 2015; van Voorst, 2015). However, social networks and community cohesion are also described to lead to problematic adaptation effects in some instances. An example is the preservation of exposure to flooding because individuals 395 refuse to leave at-risk areas as they want to stay in the network that gives them a strong sense of belonging and livelihood security (Hellman, 2015; Neolaka, 2012; Rahmayati et al., 2017). Besides this, there are authors who criticise that the often positively portrayed community-led adaptation approaches do not lead to optimal and sustainable adaptation, as actions are often implemented in a reactive ad-hoc and rather uncoordinated manner without sufficient financial means (Marfai et al., 2015; Ward et al., 2013b).


## 6. Discussion

Our review shows that there is a rich, diverse and rapidly growing body of literature analyzing Jakarta's flood problem and identifying as well as evaluating adaptation options. Looking at the discussed drivers of flooding to answer our first research question, we find that one stream of literature; i.e. publications from the fields flood models & flood mapping as well as hard 405 adaptation, predominantly frame flooding as being caused by environmental physical factors and hence as a hazard that can be controlled through engineering solutions and environmental management. Looking at the size of this body of literature measured by the numbers of publications, this can be considered the main stream of flood risk research in Jakarta. A significantly smaller body of literature composing of studies on soft and hybrid adaptation measures, including a focus on social vulnerability reduction and integrated water management, acknowledges the natural and environmental drivers of 410 flooding but also highlights the importance of socio-economic flood drivers, arguing that these drivers equally have to be considered in the consideration and design of adaptation solutions.

Research question two of this study asks which measures for risk management and adaptation are considered in the literature. Our analysis shows that there are largely two separate perspectives on suitable measures for adapting to flooding in Jakarta. 415 One follows a protection approach, identifying predominantly infrastructural measures such as dams, sea walls, water canalization or reservoir constructions as solutions to protect the city from flooding. The second one is not opposed to infrastructural measures but criticises how they are implemented and demands among others for the inclusion of soft adaptation

options to achieve comprehensive flood risk management and hybrid adaptation approaches. This literature argues that without a consideration of the root causes of social vulnerability and flood risk, hard infrastructure solutions are bound to be insufficient or even ineffective in the long-run. Likewise, these studies argue that social and environmental effects of hard protection measures need to be considered more stringently and that soft or hybrid approaches are oftentimes better-placed to create synergies with other development objectives.

Evaluating these findings from our own perspective we identify the following five points, which we argue might be helpful for advancing the debate and complementing the current perception on the adaptation solution space in Jakarta:

First, the focus on natural drivers of flooding reinforces the perception of flooding being a hazard that can be controlled by technical measures, skewing the perceived solution space towards hard adaptation measures. While it is, of course, very important to assess and understand natural drivers of flooding in Jakarta, other drivers of flood risk need to be considered with the same rigour in order to design effective adaptation options. Hard measures are – and will be – undoubtedly an important part of Jakarta's solution space. Yet, a shortfall of consideration on the anthropogenic flood drivers yields the risk to design infrastructural solutions that only address parts of the flood problem, hence risking being less effective.

Second, we find an overall lack of considering future developments in terms of both environmental as well as socio-economic changes. Surprisingly, the majority of studies that focus on flood modelling does not yet consider future changes in environmental conditions due to climate change. Some studies look at different flood event return periods as a proxy for changing environmental conditions, however, with considerable uncertainties remaining (Budiyono et al., 2015; Juliastuti et al., 2018; Kurniyaningrum et al., 2019; Liu et al., 2015; Syafalni et al., 2015; Yayuk Supomo et al., 2018). Only a few of the publications focusing on flood modelling and hard adaptation measures consider future urbanization or socio-demographic changes and their impacts as drivers of flood risk. While some incorporate future changes in land-use (Budiyono et al., 2016; Fajar Januriyadi et al., 2018; Latief et al., 2018; Mishra et al., 2018; Rafiei Emam et al., 2016; Riyando Moe et al., 2017; Shokhrukh-Mirzo Jalilov et al., 2018; Sutrisno, 2011; Takagi et al., 2016b; Vollmer et al., 2015, 2016; Ward et al., 2011b; Ward et al., 2013c), no study in the sample considers future changes in exposure due to e.g. population growth or urban development. Similarly, the publications on hard flood protection measures mostly neglect future climatic, demographic, socio-economic and land use changes when assessing the effectiveness of existing or suggested infrastructural measures for flood protection. Some authors use return periods of rainfall events (Mantasa Salve Prastica, 2018) or floods (Ajiwibowo, 2018; Indrawati et al., 2018; Nugroho et al., 2018; Wurjanto, 2018); however, without referring to potential future changes. An exception (Takagi et al., 2017) evaluates the effectiveness of planned coastal dykes using flood and subsidence projections until the year 2050. The implications of this shortcoming for the solution space are rather weighty: developed solutions – hard, soft and hybrid – that are lacking the consideration of future developments have an inherent risk of not being sustainable and effective in the long run. While they might address current challenges very well, there is a risk that dynamic changes in environmental and/or socio-economic aspects will impact their effectiveness in the future.

Third, most publications that assess and evaluate the effectiveness of hard measures for flood management in Jakarta do not consider social aspects such as the measures' impacts on social vulnerability or the acceptance of the analyzed measures. This can be an important shortcoming when evaluating the overall adequacy and success of a hard adaptation measure. For instance, resettlements, which are often a precondition for the implementation of hard adaptation measures, significantly influence communities' livelihood opportunities and social structures (Garschagen et al., 2018; Garschagen and Surtiari, 2018; Ichwatus Sholihah and Shaojun, 2018; Surtiari et al., 2017). Furthermore, authors claim that soft aspects such as e.g. risk perception and awareness, risk communication (e.g. van Voorst, 2016), behavioral and cultural factors (e.g. Bott et al., 2019; Yoga Putra et al., 2019b), collective action as well as participatory planning (e.g Sugar et al., 2013), coordination capacities (e.g. Marfai et al., 2015; Padawangi and Douglass, 2015) and law enforcement are inherently intertwined with the success of mitigation and adaptation efforts in the long run, hence calling for a rigorous consideration. The neglect of aspects such as social acceptance as well as impacts of infrastructural flood protection on local communities influences the solution space in that hard adaptation measures will be considered for flood risk management despite their potential negative impacts on social vulnerabilities. Accordingly, the solution space contains measures, which are beneficial for some groups of people while representing a threat

to other groups. While this is already alluded to in the assessed literature (e.g. Van Voorst and Hellman, 2015), this seems to have had only marginal influence on flood risk research in Jakarta until today.

Fourth, studies focusing on soft and hybrid adaptation measures converge in their critique of technocratic approaches and provide a broad variety of needs with respect to improving flood risk management, however without providing concrete
recommendations how to achieve them. Many of these studies suggest a broad range of needs to improve the current flood management approach in Jakarta. Depending on the perspective of the author(s), it is for instance advocated for more integrated and hybrid adaptation approaches (e.g. Akmalah and Grigg, 2011; David et al., 2016; Shokhrukh-Mirzo Jalilov et al., 2018) and awareness raising for and dissemination of risk and response information to foster behavioral change within the public, authorities and among urban planners (e.g. Akmalah and Grigg, 2011; Amri et al., 2017; Esteban et al., 2017; Goh, 2019;
Marfai et al., 2015; Neolaka, 2012, 2013; Nurhidayah and McIlgorm, 2019; van Voorst and Hellman, 2015; Ward et al., 2013b). Others call for regional approaches to land use and urban planning (e.g. integration of upstream and downstream, Jakarta and Jabodetabek[2] (Asdak et al., 2018; Firman et al., 2011; Goh, 2019; Mathewson, 2018; Noviandi et al., 2017) and more integrated legal and institutional frameworks as well as strong institutional bodies for disaster risk reduction and climate change adaptation (Akmalah and Grigg, 2011; Asdak et al., 2018; Firman et al., 2011; Garschagen et al., 2018; Garschagen
and Surtiari, 2018; Nurhidayah and McIlgorm, 2019; Octavianti and Charles, 2018; Ward et al., 2013b). Moreover, studies highlight a need for stronger law enforcement (e.g. Akmalah and Grigg, 2011), increased community-participation in risk management planning and decision-making (Goh, 2019) as well as in resettlements/relocation (Ichwatus Sholihah and Shaojun, 2018; Texier, 2008). However, the publications provide little concrete recommendations on how to achieve these goals: How to decide for different adaptation measures for a balanced hybrid adaptation approach? How to improve the legal and
institutional setup, which would most likely include altering current political structures and decision-making processes? How to integrate flood risk policies with the wider development agenda? How to facilitate participatory flood risk management? There are only very few publications (e.g. Amri et al., 2017; Asdak et al., 2018; Firman et al., 2011; Nurhidayah and McIlgorm, 2019) providing slightly more detailed information on the implementation and feasibility of suggested measures. While studies on soft and hybrid measures add valuable knowledge and potential options to the solution space, their lack of actionable
recommendations limit their utility. They are more difficult to consider in actual adaptation planning – especially in comparison to hard adaptation measures which often have clear requirements and quantitative assessments of their use.

Fifth, there is a lack of studies that compare multiple adaptation solutions. Apart from a few exceptions (e.g. Lin, Shaad, and Girot 2016; Fitrinitia et al., 2018) this lack applies to comparisons between different infrastructural measures as well as
between hard and soft measures. This is surprising against the background of the rich diversity of assessments of single measures – be it hard adaptation options such as levees, dams, dikes and embankments (Mardjono et al., 2018; Mardjono and Setiawan, 2018; Ogie et al., 2016b; Su et al., 2018; Sujono, 2012; Susilo et al., 2019; Suprayogi et al., 2018; Takagi et al., 2016a, 2017; Wurjanto, 2018), water channeling, retention ponds and drainage systems (Indrawati et al., 2018; Kadri, 2011; Kartolo and Kusumawati, 2017; Mahanani and Chotib, 2018; Mohajit, 2015; Nugroho et al., 2018; Sholichin et al., 2019;
Wihaji et al., 2018) or soft adaptation measures like mutual support through social networks (Guinness, 2019; Leitner and Sheppard, 2017; Padawangi and Douglass, 2015; van Voorst and Hellman, 2015), collective action in e.g. river monitoring and early warning (Bahri and W Purwantiasning, 2019; Dwirahmadi et al., 2013; Fitrinitia et al., 2018; Hellman, 2015; Padawangi and Douglass, 2015; van Voorst, 2014, 2016) or self-organization of saving groups for flood response and recovery (Dwirahmadi et al., 2013; Marfai et al., 2015; van Voorst, 2014). For the solution space, this aspect can be considered as the
most important gap. The absence of comparisons of different measures leaves decision-makers without scientific guidance in understanding the advantages and disadvantages of one adaptation solution over the another. Without comparatively considering effectiveness, social and environmental impacts as well as feasibility of different measures, it is very difficult to identify "best" adaption options and combine them into sustainable adaptation pathways.

---

[2] Jabodetabek is an acronym for the metropolitan area of Jakarta. Besides the city of Jakarta, it includes Bogor, Depok, Tangerang and Bekasi.

## 7. Conclusions and outlook

This study aimed at assessing how the solution space for flood risk reduction and climate change adaptation in Jakarta is currently being perceived, framed and evaluated in the academic literature. Learning from Jakarta, one of the cities with the highest flood-risk globally, is relevant since many other cities around the globe will be faced with similar challenges over the course of the next decades. The findings show that a focus on environmental flood drivers, numeric flood modelling and hard flood protection solutions constitutes the main centre of gravity within the current epistemic landscape of the flood risk and adaptation science on Jakarta. Soft and hybrid adaptation measures as well as potential shortcomings in hard protection approaches receive increasing, yet overall considerably less, scientific attention. While hard adaptation measures are – and will remain to be – of key importance for Jakarta to address current and future flood risk in an effective manner, the results nevertheless suggest that the identified imbalance in the current focus is problematic. If not complemented by other perspectives, the focus on hard protection bears the risk that measures which address flood symptoms are prioritized over those addressing the root causes of flood risk and the sources of social vulnerability. In addition, there is the risk that the potential of additional or complementary soft adaptation measures at different scales and implemented by different actors (state, civil society, private sector) – is not being given adequate attention in adaptation discourses at the science-policy-interface and eventually will not be used and fostered for crafting actual adaptation pathways. Hence, the findings suggest that a considerable part of the potential solution space remains to be underrepresented in the debate and not advanced with full proficiency..

Relating these findings to global research frontiers, it is striking that the above gaps in the state of science on Jakarta's flood risk and its reduction are so persistent. The literature on Jakarta has been rising sharply and Jakarta certainly belongs to the most-researched coastal high-risk cities in the world. Yet, our analysis suggests that this high potential has so far not been sufficiently used to inform and advance some of the most pressing frontiers in coastal urban risk and adaptation research: How to develop and test better approaches to model and assess future trends in socio-economic vulnerability within cities; how to evaluate different competing adaptation options in an integrative way whilst also including aspects of social acceptance and equity; how to design adaptation pathways with a mixture of hard and soft adaptation options so as to seek synergies whilst overcoming the shortcoming any isolated approach would have; how to chart and navigate transformational adaptation that shifts the political economy of risk production and the existing paradigms of adaptation away from technocratic fixes and towards the root causes of flood risk and social vulnerability.

While our findings and lessons from Jakarta cannot easily be transferred one-to-one to other risk context, recent global assessments (Oppenheimer et al., 2019) suggest the many cities might be facing similar patterns as the ones identified for Jakarta, calling for follow-up research. We therefore hope that the perspectives and questions raised in this paper are useful to inspire studies on the solution spaces in other high risk settings.

## Appendices

**Appendix A: Results from the structured literature search**

In total, 339 publications were identified by the keyword search. 14 proceedings and two publications for which no full-text version were available have been excluded. The below list shows the remaining 323 publications. All references that are marked with an asterics were included in the detailed content analysis and are cited in the paper.

*Abidin, H. Z.: Land Subsidence in Urban Areas of Indonesia: Suitability of levelling, GPS and INSAR for monitoring, GIM Int., 19, 12–15, 2005.

Abidin, H. Z., Andreas, H., Gamal, M., Gumilar, I., Napitupulu, M., Fukuda, Y., Deguchi, T., and Riawan, Y. M. & E.: Land subsidence characteristics of the Jakarta basin (Indonesia) and its relation with groundwater extraction and sea level rise, in: Groundwater Response to Changing Climate, CRC Press, 2010.

Abidin, H. Z., Andreas, H., Gumilar, I., Sidiq, T. P., and Fukuda, Y.: On the Roles of Geospatial Information for Risk Assessment of Land Subsidence in Urban Areas of Indonesia, in: Intelligent Systems for Crisis Management: Geo-information for Disaster Management (Gi4DM) 2012, edited by: Zlatanova, S., Peters, R., Dilo, A., and Scholten, H., Springer, Berlin, Heidelberg, 277–288, https://doi.org/10.1007/978-3-642-33218-0_19, 2013.

*Abidin, H. Z., Andreas, H., Gumilar, I., Yuwono, B. D., Murdohardono, D., and Supriyadi, S.: On Integration of Geodetic Observation Results for Assessment of Land Subsidence Hazard Risk in Urban Areas of Indonesia, in: IAG 150 Years, vol. 143, edited by: Rizos, C. and Willis, P., Springer International Publishing, Cham, 435–442, https://doi.org/10.1007/1345_2015_82, 2015a.

Abidin, H. Z., Andreas, H., Gumilar, I., Yuwono, B. D., Murdohardono, D., and Supriyadi, S.: On Integration of Geodetic Observation Results for Assessment of Land Subsidence Hazard Risk in Urban Areas of Indonesia, in: IAG 150 Years - Proceedings of the 2013 IAG Scientific Assembly, Postdam,Germany, 1–6 September, 2013, vol. 143, edited by: Rizos, C. and Willis, P., Springer International Publishing, 435–442, https://doi.org/10.1007/1345_2015_82, 2015b.

Abidin, H. Z., Andreas, H., Gumilar, I., and Brinkman, J. J.: Study on the risk and impacts of land subsidence in Jakarta, Proc. Int. Assoc. Hydrol. Sci., 372, 115–120, https://doi.org/10.5194/piahs-372-115-2015, 2015c.

Achmad Santosa, M., Khatarina, J., and Assegaf, R. sjarief: Indonesia, in: Climate Change Liability: Transnational Law and Practice, edited by: Brunnée, J., Rajamani, L., Lord, R., and Goldberg, S., Cambridge University Press, Cambridge, 178–205, https://doi.org/10.1017/CBO9781139084383.011, 2011.

*Aditya, M. R., Hernina, R., and Rokhmatuloh: Geographic Information System and Remote Sensing Approach with Hydrologic Rational Model for Flood Event Analysis in Jakarta, IOP Conf. Ser. Earth Environ. Sci., 98, 012008, https://doi.org/10.1088/1755-1315/98/1/012008, 2017.

*Aerts, J., Botzen, W., Bowman, M., Dircke, P., and Ward, P.: Climate Adaptation and Flood Risk in Coastal Cities, Routledge, https://doi.org/10.4324/9781849776899, 2013.

*Agustan, A., Sanjaya, H., and Ito, T.: Jakarta Land Subsidence and Inundation Vulnerability Based on SAR Data, 34th Asian Conference on Remote Sensing (ACRS) 2013, Bali, Indonesia, 2013.

*Ajiwibowo, H.: The Influence Of The Jakarta Bay Reclamation On The Surrounding Tidal Elevation And Tidal Current, Int. J. GEOMATE, 15, https://doi.org/10.21660/2018.48.22773, 2018.

*Akmalah, E. and Grigg, N. S.: Jakarta flooding: systems study of socio-technical forces, Water Int., 36, 733–747, https://doi.org/10.1080/02508060.2011.610729, 2011.

Alatas, I. F.: Becoming Indonesians: The Bā 'Alawī in the Interstices of the Nation, Welt Islams, 51, 45–108, https://doi.org/10.1163/157006011X556120, 2011.

Alfida, A., Maryam, S., and Rianti, F.: Information in the Age of Misinformation: Counteracting the Problems of Online Radicalization with Digital Literacy, Libr. Philos. Pract. E-J., 2019.

Álvarez, J. C. and Resosudarmo, B. P.: The cost of floods in developing countries' megacities: A hedonic price analysis of the Jakarta housing market, Indonesia, Departmental Working Papers, The Australian National University, Arndt-Corden Department of Economics, 2018.

*Amri, A., Bird, D. K., Ronan, K., Haynes, K., and Towers, B.: Disaster risk reduction education in Indonesia: challenges and recommendations for scaling up, Nat. Hazards Earth Syst. Sci., 17, 595–612, https://doi.org/10.5194/nhess-17-595-2017, 2017.

*Andreas, H., Usriyah, Zainal Abidin, H., and Anggreni Sarsito, D.: Tidal inundation ("Rob") investigation using time series of high resolution satellite image data and from institu measurements along northern coast of Java (Pantura), IOP Conf. Ser. Earth Environ. Sci., 71, 012005, https://doi.org/10.1088/1755-1315/71/1/012005, 2017.

*Andreas, H., Zainal Abidin, H., Pradipta, D., Anggreni Sarsito, D., and Gumilar, I.: Insight look the subsidence impact to infrastructures in Jakarta and Semarang area; Key for adaptation and mitigation, MATEC Web Conf., 147, 08001, https://doi.org/10.1051/matecconf/201814708001, 2018.

*Andreas, H., Abidin, H. Z., Sarsito, D. A., and Pradipta, D.: Determining the initial time of anthropogenic subsidence in urban area of Indonesia, IOP Conf. Ser. Earth Environ. Sci., 389, 012034, https://doi.org/10.1088/1755-1315/389/1/012034, 2019a.

Andreas, H., Abidin, H. Z., Sarsito, D. A., Meilano, I., and Susilo: Investigating the tectonic influence to the anthropogenic subsidence along northern coast of Java Island Indonesia using GNSS data sets, E3S Web Conf., 94, 04005, https://doi.org/10.1051/e3sconf/20199404005, 2019b.

Andromeda, Z. I., Randi, C., Nazhifah, K., Syahputra, R., Iskandarsyah, and Septyandy, M. R.: Efforts to Improve Community Awareness Towards the Potential of a Great Earthquake Which Threats Jakarta Based on Geographic Information and 3D
Simulation Systems in Matraman District, East Jakarta: 1st International Conference on Geoscience, ICoGeS 2018, IOP Conf. Ser. Earth Environ. Sci., 279, https://doi.org/10.1088/1755-1315/279/1/012004, 2019.

Anggraheni, E., Sutjiningsih, D., Widyoko, J., and Hidayah, B.: Potential impact of sub-urban development on the surface runoff estimations (a case study at Upper Ciliwung watershed): 12th Thai Society of Agricultural Engineering International Conference, TSAE 2019, IOP Conf. Ser. Earth Environ. Sci., 301, https://doi.org/10.1088/1755-1315/301/1/012002, 2019.

Anggraini, D., Odang, S., and Mustadjab, I.: Proposed Design of Bicycle Lanes Around Jakarta's East Flood Canal, in: Sustainable Building and Built Environments to Mitigate Climate Change in the Tropics: Conceptual and Practical Approaches, edited by: Karyono, T. H., Vale, R., and Vale, B., Springer International Publishing, Cham, 149–163, https://doi.org/10.1007/978-3-319-49601-6_11, 2017.

*Anindita, A. P., Laksono, P., and Nugraha, I. G. B. B.: Dam water level prediction system utilizing Artificial Neural Network
Back Propagation: Case study: Ciliwung watershed, Katulampa Dam, in: 2016 International Conference on ICT For Smart Society (ICISS), 2016 International Conference on ICT For Smart Society (ICISS), Surabaya, Indonesia, 16–21, https://doi.org/10.1109/ICTSS.2016.7792862, 2016.

Anugrahadi, A., Yuda, H. F., and Triany, N.: The Jakarta Flood Disaster Mapping January 2020, Int. J. Adv. Sci. Technol., 29, 9, 2020.

Arifin, H. S. and Nakagoshi, N.: Landscape ecology and urban biodiversity in tropical Indonesian cities, Landsc. Ecol. Eng., 7, 33–43, https://doi.org/10.1007/s11355-010-0145-9, 2011.

Asdak, C., Supian, S., and Subiyanto: Watershed management strategies for flood mitigation: A case study of Jakarta's flooding, Weather Clim. Extrem., 21, 117–122, https://doi.org/10.1016/j.wace.2018.08.002, 2018.

*Asmadin, Siregar, V. P., Sofian, I., Jaya, I., and Wijanarto, A. B.: Feature extraction of coastal surface inundation via water 630 index algorithms using multispectral satellite on North Jakarta, IOP Conf. Ser. Earth Environ. Sci., 176, 1–10, https://doi.org/10.1088/1755-1315/176/1/012032, 2018.

Augustine Adibroto, T., Wijayanti, P., Pratama Adhi, R., and Nugroho, R.: Preliminary study on socio-economic aspect towards Jakarta climate resilient (case study: Cengkareng District, West Jakarta and Penjaringan District, North Jakarta), IOP Conf. Ser. Earth Environ. Sci., 314, 012035, https://doi.org/10.1088/1755-1315/314/1/012035, 2019.

*Badriana, M. R., Bachtiar, H., Adytia, D., Sembiring, L., Andonowati, and van Groesen, E.: Wave run-up of a possible Anak-Krakatau tsunami on planned and optimized Jakarta Sea Dike, INTERNATIONAL SYMPOSIUM ON EARTH HAZARD AND DISASTER MITIGATION (ISEDM) 2016: The 6th Annual Symposium on Earthquake and Related Geohazard Research for Disaster Risk Reduction, Bandung, Indonesia, https://doi.org/10.1063/1.4987103, 2017.

*Bahri, S. and W Purwantiasning, A.: An Application of Microcontroller for Flood Hazard Early Warning System to Create 640 Friendly City, J. Phys. Conf. Ser., 1376, 012016, https://doi.org/10.1088/1742-6596/1376/1/012016, 2019.

*Bahtiar, S., Chuai-Aree, S., and Busaman, A.: A Numerical Algorithm and Visualization Software for Flood Simulation in Urban Area: A Case Study of West Jakarta, Indonesia, 12, 147–153, 2018.

Bajraghosa, T.: Komik mandiri in Yogyakarta; local values representation in independent comics, Humanit. Arts Soc. Sci. Stud., 19, 387–404, 2019.

*Batubara, B., Kooy, M., and Zwarteveen, M.: Uneven Urbanisation: Connecting Flows of Water to Flows of Labour and Capital Through Jakarta's Flood Infrastructure, Antipode, 50, 1186–1205, https://doi.org/10.1111/anti.12401, 2018.

Birkmann, J., Cutter, S. L., Rothman, D. S., Welle, T., Garschagen, M., van Ruijven, B., O'Neill, B., Preston, B. L., Kienberger, S., Cardona, O. D., Siagian, T., Hidayati, D., Setiadi, N., Binder, C. R., Hughes, B., and Pulwarty, R.: Scenarios for vulnerability: opportunities and constraints in the context of climate change and disaster risk, Clim. Change, 133, 53–68, 650 https://doi.org/10.1007/s10584-013-0913-2, 2015.

*Bott, L.-M., Ankel, L., and Braun, B.: Adaptive neighborhoods: The interrelation of urban form, social capital, and responses to coastal hazards in Jakarta, Geoforum, 106, 202–213, https://doi.org/10.1016/j.geoforum.2019.08.016, 2019.

Bucx, T. H. M., van Ruiten, C. J. M., Erkens, G., and de Lange, G.: An integrated assessment framework for land subsidence in delta cities, in: Proceedings of the International Association of Hydrological Sciences, Prevention and mitigation of natural 655 and anthropogenic hazards due to land subsidence - Ninth International Symposium on Land Subsidence (NISOLS), Nagoya, Japan, 15–19 November 2015, 485–491, https://doi.org/10.5194/piahs-372-485-2015, 2015.

*Budiyono, Y., Aerts, J., Brinkman, J., Marfai, M. A., and Ward, P.: Flood risk assessment for delta mega-cities: a case study of Jakarta, Nat. Hazards, 75, 389–413, https://doi.org/10.1007/s11069-014-1327-9, 2015.

*Budiyono, Y., Aerts, J. C. J. H., Tollenaar, D., and Ward, P. J.: River flood risk in Jakarta under scenarios of future change,
Nat. Hazards Earth Syst. Sci., 16, 757–774, https://doi.org/10.5194/nhess-16-757-2016, 2016.

Caljouw, M., Nas, P. J. M., and Pratiwo: Flooding in Jakarta: Towards a blue city with improved water management, Bijdr.
Tot Taal- Land- En Volkenkd., 161, 454–484, 2005.

*Chaussard, E., Amelung, F., Abidin, H., and Hong, S.-H.: Sinking cities in Indonesia: ALOS PALSAR detects rapid
subsidence due to groundwater and gas extraction, Remote Sens. Environ., 128, 150–161,
https://doi.org/10.1016/j.rse.2012.10.015, 2013.

*Colven, E.: Understanding the Allure of Big Infrastructure: Jakarta's Great Garuda Sea Wall Project, Water Altern., 10, 250–
264, 2017.

Comber, A. J.: Improving land cover classification using input variables derived from a geographically weighted principal
components analysis, ISPRS J. Photogramm. Remote Sens., 14, 2016.

*Costa, D., Burlando, P., and Priadi, C.: The importance of integrated solutions to flooding and water quality problems in the
tropical megacity of Jakarta, Sustain. Cities Soc., 20, 199–209, https://doi.org/10.1016/j.scs.2015.09.009, 2016.

Daksiya, V., Su, H. T., Chang, Y. H., and Lo, E. Y. M.: Incorporating socio-economic effects and uncertain rainfall in flood
mitigation decision using MCDA, Nat. Hazards, 87, 515–531, https://doi.org/10.1007/s11069-017-2774-x, 2017.

Dalimunthe, S. A.: Rural Indonesian Insight on Mass Media Role in Reducing Climate Change Risk, in: Handbook of Climate
Change Communication: Vol. 2: Practice of Climate Change Communication, edited by: Leal Filho, W., Manolas, E., Azul,
A. M., Azeiteiro, U. M., and McGhie, H., Springer International Publishing, Cham, 61–67, https://doi.org/10.1007/978-3-319-
70066-3_5, 2018.

Damayanti, A. and Dwiputra, N. A.: Flood Exposure of Settlement Areas in Bekasi City, 7, 2018.

*David, C. G., Schulz, N., and Schlurmann, T.: Assessing the Application Potential of Selected Ecosystem-Based, Low-Regret
Coastal Protection Measures, in: Ecosystem-Based Disaster Risk Reduction and Adaptation in Practice, vol. 42, edited by:
Renaud, F. G., Sudmeier-Rieux, K., Estrella, M., and Nehren, U., Springer International Publishing, Cham, 457–482,
https://doi.org/10.1007/978-3-319-43633-3_20, 2016.

Deppman, H.-C.: Made in Taiwan: An Analysis of Meteor Garden as an Asian Idol Drama, in: TV China, edited by: Zhu, Y.
and Berry, C., Indiana University Press, Bloomington, 2008.

Dewandaru, A., Supriana, S. I., and Akbar, S.: Event-Oriented Map Extraction From Web News Portal : Binary Map Case
Study on Diphteria Outbreak and Flood in Jakarta, 5th International Conference on Advanced Informatics: Concept Theory
and Applications (ICAICTA), Krabi, Thailand, 72–77, https://doi.org/10.1109/ICAICTA.2018.8541345, 2018.

Diposaptono, S., Pratikto, W. A., and Mano, A.: Flood in jakarta - lessons learnt from the 2002 flood, in: Asian and Pacific
Coasts 2003, WORLD SCIENTIFIC, 1–8, https://doi.org/10.1142/9789812703040_0006, 2004.

*Douglass, M.: Globalization, Mega-projects and the Environment: Urban Form and Water in Jakarta, Environ. Urban. ASIA,
1, 45–65, https://doi.org/10.1177/097542530900100105, 2010.

*Dovey, K., Cook, B., and Achmadi, A.: Contested riverscapes in Jakarta: flooding, forced eviction and urban image, Space
Polity, 23, 265–282, https://doi.org/10.1080/13562576.2019.1667764, 2019.

Dsikowitzky, L., Schäfer, L., Dwiyitno, Ariyani, F., Irianto, H. E., and Schwarzbauer, J.: Evidence of massive river pollution in the tropical megacity Jakarta as indicated by faecal steroid occurrence and the seasonal flushing out into the coastal ecosystem, Environ. Chem. Lett., 15, 703–708, https://doi.org/10.1007/s10311-017-0641-3, 2017.

Dulal, H. B.: Cities in Asia: how are they adapting to climate change?, J. Environ. Stud. Sci., 9, 13–24, https://doi.org/10.1007/s13412-018-0534-1, 2019.

*Dwirahmadi, F., Rutherford, S., Urlich, W., and Chu, C.: Linking disaster risk reduction and climate change adaptation: a good practice project in Jakarta, Indonesia, in: Climate Adaptation Futures, edited by: Palutikof, J., Boulter, S. L., Ash, A. J., Stafford Smith, M., Parry, M., Waschka, M., and Guitart, D., John Wiley & Sons, Ltd., 362–370, 2013.

Dwirahmadi, F., Rutherford, S., Phung, and Chu, C.: Understanding the Operational Concept of a Flood-Resilient Urban Community in Jakarta, Indonesia, from the Perspectives of Disaster Risk Reduction, Climate Change Adaptation and Development Agencies, Int. J. Environ. Res. Public. Health, 16, 3993, https://doi.org/10.3390/ijerph16203993, 2019.

Edinger, E. and Browne, D.: Continental Seas of Western Indonesia, in: Seas at the Millennium - An Environmental Evaluation, vol. Vol. 2, edited by: Sheppard, C., Elsevier Science Ltd., 381–404, 2000.

Eilander, D., Trambauer, P., Wagemaker, J., and van Loenen, A.: Harvesting Social Media for Generation of Near Real-time Flood Maps, Procedia Eng., 154, 176–183, https://doi.org/10.1016/j.proeng.2016.07.441, 2016.

El-Abbas, A. A. and Shedid, S. A.: Experimental Investigation of the Feasibility of Steam/Chemical Steam Flooding Processes through Horizontal Wells, SPE Asia Pacific Oil and Gas Conference and Exhibition, https://doi.org/10.2118/68767-MS, 2001.

Erkens, G., Bucx, T., Dam, R., de Lange, G., and Lambert, J.: Sinking coastal cities, in: Proceedings of the International Association of Hydrological Sciences, Prevention and mitigation of natural and anthropogenic hazards due to land subsidence - Ninth International Symposium on Land Subsidence (NISOLS), Nagoya, Japan, 15–19 November 2015, 189–198, https://doi.org/10.5194/piahs-372-189-2015, 2015.

*Esteban, M., Takagi, H., Mikami, T., Aprilia, A., Fujii, D., Kurobe, S., and Utama, N. A.: Awareness of coastal floods in impoverished subsiding coastal communities in Jakarta: Tsunamis, typhoon storm surges and dyke-induced tsunamis, Int. J. Disaster Risk Reduct., 23, 70–79, https://doi.org/10.1016/j.ijdrr.2017.04.007, 2017.

Esteban, M., Jamero, Ma. L., Nurse, L., Yamamoto, L., Takagi, H., Thao, N. D., Mikami, T., Kench, P., Onuki, M., Nellas, A., Crichton, R., Valenzuela, V. P., Chadwick, C., Avelino, J. E., Tan, N., and Shibayama, T.: Adaptation to sea level rise on low coral islands: Lessons from recent events, Ocean Coast. Manag., 168, 35–40, https://doi.org/10.1016/j.ocecoaman.2018.10.031, 2019.

Fachrul, M. F., Rinanti, A., and Hendrawan, D. I.: Assessing Water Quality by Bioindicator Inat Pluit Reservoir in Jakarta Urban Area, Indonesia, Int. J. Adv. Sci. Technol., 29, 9, 2020.

*Faedlulloh, D., Prasetyanti, R., and Irawan, B.: Kampung versus Climate Change: The Dynamics of Community Empowerment through the Climate Village Program (ProKlim), J. Phys. Conf. Ser., 1424, 8, 2019.

*Fajar Januriyadi, N., Kazama, S., Riyando Moe, I., and Kure, S.: Evaluation of future flood risk in Asian megacities: a case study of Jakarta, Hydrol. Res. Lett., 12, 14–22, https://doi.org/10.3178/hrl.12.14, 2018.

*Farid, M., Mano, A., and Udo, K.: Distributed flood model for urbanization assessment in a limited-gauged river basin, RIVER BASIN MANAGEMENT 2011, Riverside, California, USA, 83–94, https://doi.org/10.2495/RM110081, 2011.

*Farid, M., Mano, A., Udo, K., Water Resources Engineering Research Group, Institut Teknologi Bandung, Jalan Ganesha 10 Bandung 40132, Indonesia, and Disaster Control Research Center, Graduate School of Engineering, Tohoku University, 6-6-11 Aoba Aramaki Aoba-ku, Sendai 980-8579, Japan: Urban Flood Inundation Model for High Density Building Area, J. Disaster Res., 7, 554–559, https://doi.org/10.20965/jdr.2012.p0554, 2012.

Farid, M., Harumi Pusparani, H., Syahril Badri Kusuma, M., and Natasaputra, S.: Study on effectiveness of flood control based
on risk level: case study of Kampung Melayu Village and Bukit Duri Village, MATEC Web Conf., 101, https://doi.org/10.1051/matecconf/201710105003, 2017.

Febrianto, H., Fariza, A., and Hasim, J.: Urban flood risk mapping using analytic hierarchy process and natural break classification (Case study: Surabaya, East Java, Indonesia), 2016 International Conference on Knowledge Creation and Intelligent Computing (KCIC), Manado, Indonesia, https://doi.org/10.1109/KCIC.2016.7883639, 2016.

Fernando, M. and Gaol, F. L.: Designing flood early warning system using internet of things, in: AIP Conference Proceedings, 020020, https://doi.org/10.1063/1.5080039, 2018.

*Firman, T., Surbakti, I. M., Idroes, I. C., and Simarmata, H. A.: Potential climate-change related vulnerabilities in Jakarta: Challenges and current status, Habitat Int., 35, 372–378, https://doi.org/10.1016/j.habitatint.2010.11.011, 2011.

*Fitrinitia, I. S., Junadi, P., Sutanto, E., Nugroho, D. A., Zubair, A., and Suyanti, E.: Local adaptive capacity as an alternative
approach in dealing with hydrometeorological risk at Depok Peri-Urban City, IOP Conf. Ser. Earth Environ. Sci., 129, https://doi.org/10.1088/1755-1315/129/1/012015, 2018.

*Formánek, A., Silasari, R., Kusuma, M. S. B., and Kardhana, H.: Two-Dimensional Model of Ciliwung River Flood in DKI Jakarta for Development of the Regional Flood Index Map, J. Eng. Technol. Sci., 45, 307–325, https://doi.org/10.5614/j.eng.technol.sci.2013.45.3.7, 2013.

Furlong, K. and Kooy, M.: Worlding Water Supply: Thinking Beyond the Network in Jakarta, Int. J. Urban Reg. Res., 41, 888–903, https://doi.org/10.1111/1468-2427.12582, 2017.

Gambolati, G. and Teatini, P.: Geomechanics of subsurface water withdrawal and injection, Water Resour. Res., 51, 3922–3955, https://doi.org/10.1002/2014WR016841, 2015.

*Garschagen, M. and Surtiari, G. A. K.: Hochwasser in Jakarta – zwischen steigendem Risiko und umstrittenen
Anpassungsmaßnahmen, Geogr. Rundsch., 6, 2018.

*Garschagen, M., Surtiari, G., and Harb, M.: Is Jakarta's New Flood Risk Reduction Strategy Transformational?, Sustainability, 10, 2934, https://doi.org/10.3390/su10082934, 2018.

Gharbi, R. B. C.: A Knowledge-Based System for Optimal Economic Design of Improved Recovery Processes, SPE Asia Pacific Oil and Gas Conference and Exhibition, https://doi.org/10.2118/68765-MS, 2001.

Giménez, C., Thompson, R. B., Prieto, M. H., Suárez-Rey, E., Padilla, F. M., and Gallardo, M.: Adaptation of the VegSyst model to outdoor conditions for leafy vegetables and processing tomato, Agric. Syst., 171, 51–64, 2019.

van Ginkel, K. C. H., Hoekstra, A. Y., Buurman, J., and Hogeboom, R. J.: Urban Water Security Dashboard: Systems Approach to Characterizing the Water Security of Cities, J. Water Resour. Plan. Manag., 144, 04018075, https://doi.org/10.1061/(ASCE)WR.1943-5452.0000997, 2018.

Goenawan, R. D., Ridwan, R., Sadly, M., Sudinda, T., Kudsy, M., Seto, T. H., and Harsoyo, B.: Experimental Assessment of Integrated Technology Application Used to Rain (WM4RR) & Floods Reduction (AR-DWIS) in Jakarta, Procedia Eng., 125, 270–276, https://doi.org/10.1016/j.proeng.2015.11.039, 2015.

*Goh, K.: Urban Waterscapes: The Hydro-Politics of Flooding in a Sinking City: Urban Waterscapes, Int. J. Urban Reg. Res., 43, 250–272, https://doi.org/10.1111/1468-2427.12756, 2019.

*Guinness, P.: Managing Risk in Uncertain Times, Ethnos, 85, 423–434, https://doi.org/10.1080/00141844.2018.1543341, 2019.

Hakim, R. T. and Kusumastuti, R. D.: A Model to Determine Relief Warehouse Location in East Jakarta using the Analytic Hierarchy Process, Int. J. Technol., 9, 1405–1414, https://doi.org/10.14716/ijtech.v9i7.1596, 2018.

Hanan, D. and Koesasi, B.: Betawi Moderen: Songs and Films of Benyamin S from Jakarta in the 1970s—Further Dimensions 775     of Indonesian Popular Culture, Indonesia, 35–76, https://doi.org/10.5728/indonesia.91.0035, 2011.

Hartono, D. M., Novita, E., Gusniani, I., and Oriza, I. I. D.: THE ROLE OF WATER SUPPLY AND SANITATION DURING FLOODS: CASE STUDY OF FLOOD DISASTER IN FIVE REGIONS OF JAKARTA, Int. J. Technol., 1, 29–37, 2010.

*Hellman, J.: Living with floods and coping with vulnerability, Disaster Prev. Manag. Int. J., 24, 468–483, https://doi.org/10.1108/DPM-04-2014-0061, 2015.

Hendarti, R., Katarina, W., and Wangidjaja, W.: The study of the application of crystalline silicone solar cell type for a temporary flood camp, in: IOP Conference Series: Earth and Environmental Science, IOP Conference Series Earth and Environmental Science, 012040, https://doi.org/10.1088/1755-1315/109/1/012040, 2017.

Herbeck, J. and Flitner, M.: Infrastructuring coastal futures: Key trajectories in Southeast Asian megacities, ERDE, https://doi.org/10.12854/erde-2019-451, 2019.

*Hermawan, E., Ruchjana, B. N., Abdullah, A. S., Jaya, I. G. N. M., Sipayung, S. B., and Rustiana, S.: Development of the statistical ARIMA model: an application for predicting the upcoming of MJO index, J. Phys. Conf. Ser., 893, 012019, https://doi.org/10.1088/1742-6596/893/1/012019, 2017.

Herring, S. C., Hoerling, M. P., Kossin, J. P., Peterson, T. C., and Stott, P. A.: Introduction to Explaining Extreme Events of 2014 from a Climate Perspective, Bull. Am. Meteorol. Soc., 96, S1–S4, https://doi.org/10.1175/BAMS-D-15-00157.1, 2015.

Hidayat, M. and Yunus, U.: The entrepreneurship learning in industrial 4.0 era (case study in indonesian college), J. Entrep. Educ., 22, 2019.

Hoes, O., Strijker, J., and Schuurmans, W.: Going Dutch, Int. Water Power Dam Constr., 57, 40–44, 2005.

Holderness, T. and Turpin, E.: From Social Media to GeoSocial Intelligence: Crowdsourcing Civic Co-management for Flood Response in Jakarta, Indonesia, in: Social Media for Government Services, edited by: Nepal, S., Paris, C., and Georgakopoulos, 795     D., Springer International Publishing, Cham, 115–133, https://doi.org/10.1007/978-3-319-27237-5_6, 2015.

Huhn, M., Zamani, N. P., von Juterzenka, K., and Lenz, M.: Food availability in an anthropogenically impacted habitat determines tolerance to hypoxia in the Asian green mussel Perna viridis, Mar. Biol., 163, 15, https://doi.org/10.1007/s00227-015-2786-6, 2016.

Huliana, W. and Ellisa, E.: Coworking Space and Cluster Spatial Relations in the Context of Jakarta City Spatial Structure: 2nd International Conference on Smart City Innovation, IOP Conf. Ser. Earth Environ. Sci., 396, https://doi.org/10.1088/1755-1315/396/1/012015, 2019.

*Hurford, A. P., Maksimović, C., and Leitão, J. P.: Urban pluvial flooding in Jakarta: applying state-of-the-art technology in a data scarce environment., Water Sci. Technol. J. Int. Assoc. Water Pollut. Res., 62, 2246–2255, https://doi.org/10.2166/wst.2010.485, 2010.

*Ichwatus Sholihah, P. and Shaojun, C.: Impoverishment of induced displacement and resettlement (DIDR) slum eviction development in Jakarta Indonesia, Int. J. Urban Sustain. Dev., 10, 263–278, https://doi.org/10.1080/19463138.2018.1534737, 2018.

Ignatius, S., Soeryantono, H., Anggraheni, E., and Sutjiningsih, D.: Analysis of flood inundation in North Sunter on the North Sunter Polder system performance, Mater. Sci. Eng., 12, 2019.

*Indrawati, D., Hadihardaja, I. K., Bagus Adityawan, M., Pahrizal, S. F., and Taufik, F.: Diversion Canal to Decrease Flooding (Case Study: Kebon Jati-Kalibata Segment, Ciliwung River Basin), MATEC Web Conf., 147, 5, https://doi.org/10.1051/matecconf/201814703006, 2018.

Intarti, Y. R., Fitrinitia, I. S., Widyanto, M. R., and Simarmata, H. A.: Towards Disaster Management in Indonesia Case Studies: Jakarta and Yogyakarta Provinces, Int. J. Disaster Recovery Bus. Contin., 4, 11–22, https://doi.org/10.14257/ijdrbc.2013.4.02, 2013.

Irawan, D. E., Priyambodho, A., Rachmi, C. N., and Wibowo, D. M.: Looking at groundwater research landscape of Jakarta Basin for better water management, J. Phys. Conf. Ser., 877, 012033, https://doi.org/10.1088/1742-6596/877/1/012033, 2017.

Ischak, M., Setioko, B., and Nurgandarum, D.: Socio spatial adaptation as a resilience form of native unplanned settlement in confrontation with new planned settlement development pressure (case study: enclave native settlement in Serpong, Tangerang), IOP Conf. Ser. Earth Environ. Sci., 99, 14, https://doi.org/10.1088/1755-1315/99/1/012009, 2017.

Iskandar, D. and Sugandi, D.: Flood mitigation efforts in the Capital Region of Jakarta, Int. J. Conserv. Sci., 6, 685–696, 2015.

Istiani, and Muetia, R.: The Association of Traits Personality and Pro-Social Behavior Among Volunteers in Jakarta, Adv. Sci. Lett., 22, 1738–1741, https://doi.org/10.1166/asl.2016.6752, 2016.

Jamil, S.: Climate-proofing a concrete island: improving state and societal climate adaptation capacities in Singapore, in: Adaptation to Climate Change in Asia, edited by: Vachani, S. and Usmani, J., Edward Elgar Publishing, 73–92, 2014.

*Jati, M. I. H., Suroso, and Santoso, P. B.: Prediction of flood areas using the logistic regression method (case study of the provinces Banten, DKI Jakarta, and West Java), J. Phys. Conf. Ser., 1367, 012087, https://doi.org/10.1088/1742-6596/1367/1/012087, 2019.

Jossé, G. and Scherhag, K.: Business continuity management in the destination – proactive recognition, assessment and managing of disruptions, in: Destination Resilience, Routledge, 2018.

*Juliastuti, Arumsari, P., and Setyandito, O.: Spatial Data and Catchment Discretization for Assessment Coastal Urban Drainage Performance Using GIS and MIKE URBAN-SWMM, IOP Conf. Ser. Earth Environ. Sci., 195, 1–9, https://doi.org/10.1088/1755-1315/195/1/012018, 2018.

*Kadri, T.: Flood defense in Bekasi City, Indonesia, in: Flood Recovery, Innovation and Response I, vol. I, edited by: Proverbs,
D., Brebbia, C. A., and Penning-Rowsell, E., WIT Press, 133–138, 2008.

*Kadri, T.: Lakes potency to reduce overflow discharge in the Sunter river area, Jakarta, WATER RESOURCES
MANAGEMENT 2011, Riverside, California, USA, 641–645, https://doi.org/10.2495/WRM110571, 2011.

*Kartolo, J. and Kusumawati, E.: Feasibility study of rainwater harvesting for domestic use (Case study: West Jakarta rainfall
data), in: AIP Conference Proceedings, PROCEEDINGS OF THE 3RD INTERNATIONAL CONFERENCE ON
CONSTRUCTION AND BUILDING ENGINEERING (ICONBUILD) 2017: Smart Construction Towards Global
Challenges, Palembang, Indonesia, 100008, https://doi.org/10.1063/1.5011618, 2017.

Karyono, T. H. and Bachtiar, F.: Adapting City for Frequent Floods: A Case Study of Jakarta, Indonesia, in: Sustainable
Building and Built Environments to Mitigate Climate Change in the Tropics: Conceptual and Practical Approaches, edited by:
Karyono, T. H., Vale, R., and Vale, B., Springer International Publishing, Cham, 103–111, https://doi.org/10.1007/978-3-319-
49601-6_8, 2017.

Karyono, T. H., Heryanto, S., and Faridah, I.: Air conditioning and the neutral temperature of the Indonesian university
students, Archit. Sci. Rev., 58, 174–183, https://doi.org/10.1080/00038628.2014.1002828, 2015a.

Karyono, T. H., Sri, E., Sulistiawan, J. G., and Triswanti, Y.: Thermal Comfort Studies in Naturally Ventilated Buildings in
Jakarta, Indonesia, Buildings, 5, 917–932, https://doi.org/10.3390/buildings5030917, 2015b.

Karyono, T. H., Melyan, N. H., Salsa, S. Y., and Fariz, E.: Flood Responsive Design of the Low-Income Settlements in
Kampung Melayu, Jakarta, Indonesia, in: Sustainable Building and Built Environments to Mitigate Climate Change in the
Tropics. Conceptual and Practical Approaches, edited by: Karyono, T. H., Vale, R., and Vale, B., Springer International
Publishing, 2017a.

Karyono, T. H., Burhanudin, D., and Timothi, B.: Sustainable Fishing Settlement in Muara Angke, North Jakarta, in:
Sustainable Building and Built Environments to Mitigate CLimate Change in the Tropics. Conceptual and Practical
Approaches, edited by: Karyono, T. H., Vale, R., and Vale, B., Springer International Publishing, 2017b.

Katsuhama, Y. and Grigg, N. S.: Capacity building for flood management systems: a conceptual model and case studies, Water
Int., 35, 763–778, https://doi.org/10.1080/02508060.2010.533348, 2010.

Kautsar, L. H. R., Waryono, T., and Sobirin: The return of "Gasoline station-park" status into green-open space in DKI Jakarta
Province, AIP Conf. Proc., 1862, 030197, https://doi.org/10.1063/1.4991301, 2017.

Kayane, Y.: Water Privatization in the Capital City of Indonesia and Struggles over Policy:, Jpn. J. Southeast Asian Stud., 51,
139–161, https://doi.org/10.20495/tak.51.1_139, 2013.

*Kholil and Dewi, I. K.: Evaluation of Land Use Change in the Upstream of Ciliwung Watershed to Ensure Sustainability of
Water Resources, Asian J. Water Environ. Pollut., 12, 11–19, 2014.

Koncara, R. M. P., Tiarasari, R., and Pratiwi, W. D.: Transforming Shell and Society Elements in Human Settlements for
Sustainable Tourism Development: Setu Babakan, South Jakarta, Indonesia, IOP Conf. Ser. Earth Environ. Sci., 152, 012029,
https://doi.org/10.1088/1755-1315/152/1/012029, 2018.

Konner, M.: Trauma, Adaptation, and Resilience: A Cross-Cultural and Evolutionary Perspective, in: Understanding Trauma,
edited by: Kirmayer, L. J., Lemelson, R., and Barad, M., Cambridge University Press, Cambridge, 300–338,
https://doi.org/10.1017/CBO9780511500008.021, 2007.

*Koudogbo, F. N., Duro, J., Arnaud, A., Bally, P., Abidin, H. Z., and Andreas, H.: Combined X- and L-band PSI analyses for assessment of land subsidence in Jakarta, in: Proceedings Volume 8531, Remote Sensing for Agriculture, Ecosystems, and Hydrology XIV, SPIE Remote Sensing, Edinburgh, United Kingdom, 853107, https://doi.org/10.1117/12.974821, 2012.

Kurniasari, P., Gabe, R. T., and Adianto, J.: Spatial extension as a housing strategy in kampung kota: 8th Annual International Conference 2018 on Science and Engineering, AIC-SE 2018, IOP Conf. Ser. Mater. Sci. Eng., 523, https://doi.org/10.1088/1757-899X/523/1/012049, 2019.

*Kurniawan, V.: Distribution fitting on rainfall data in Jakarta, Mater. Sci. Eng., 9, 2019.

*Kurniyaningrum, E., Limantara, L. M., Suhartanto, E., and Sisinggih, D.: Development Of Flood Early Warning System Based On The Geoinformatics System In The Krukut River, Jakarta, Indonesia, Int. J. Civ. Eng. Technol., 10, 1325–1335, 880 2019a.

Kurniyaningrum, E., Limantara, L. M., Suhartanto, E., and Sisinggih, D.: Sensitivity of flow depth inundation based on the micro-scale topography in Krukut River, Jakarta, Indonesia, Int. J. Civ. Eng. Technol., 10, 697–706, 2019b.

Kusumo, A. N. L., Reckien, D., and Verplanke, J.: Utilising volunteered geographic information to assess resident's flood evacuation shelters. Case study: Jakarta, Appl. Geogr., 88, 174–185, https://doi.org/10.1016/j.apgeog.2017.07.002, 2017.

Lamond, J. E.: Balancing Flood Risk and Water Scarcity of the Asian Delta Regions, in: Water Resources in the Built Environment, John Wiley & Sons, Ltd, 389–400, https://doi.org/10.1002/9781118809167.ch28, 2014.

*Latief, H., Putri, M. R., Hanifah, F., Afifah, I. N., Fadli, M., and Ismoyo, D. O.: Coastal Hazard Assessment in Northern part of Jakarta, Procedia Eng., 212, 1279–1286, https://doi.org/10.1016/j.proeng.2018.01.165, 2018.

*Latifah, A. L. and Setiawan, I.: Comparing deterministic and geostatistical methods for spatial rainfall distribution in Jakarta 890 area, in: 2014 2nd International Conference on Technology, Informatics, Management, Engineering Environment, 2014 2nd International Conference on Technology, Informatics, Management, Engineering Environment, 40–45, https://doi.org/10.1109/TIME-E.2014.7011589, 2014.

Lavigne, F., Texier, P., and Fort, M.: Réduction des risques d'inondation à Jakarta : de la nécessaire intégration d'une approche sociale et communautaire dans la réduction des risques de catastrophe (Reducing flood risk in Jakarta : integrating social and 895 community approaches in the reduction of disaster risk), Bull. Assoc. Géographes Fr., 87, 551–570, https://doi.org/10.3406/bagf.2010.8197, 2010.

Leelawat, N., Muhari, A., Srivichai, M., Suppasri, A., Imamura, F., and Bricker, J. D.: Preference for Information During Flood Disasters: A Study of Thailand and Indonesia, in: Sustainable Future for Human Security: Society, Cities and Governance, edited by: McLellan, B., Springer, Singapore, 335–349, https://doi.org/10.1007/978-981-10-5433-4_23, 2018.

*Leitner, H. and Sheppard, E.: From Kampungs to Condos? Contested accumulations through displacement in Jakarta, Environ. Plan. A, 0, 1–20, https://doi.org/10.1177/0308518X17709279, 2017.

*Lin, E., Shaad, K., and Girot, C.: Developing river rehabilitation scenarios by integrating landscape and hydrodynamic modeling for the Ciliwung River in Jakarta, Indonesia, Sustain. Cities Soc., 20, 180–198, https://doi.org/10.1016/j.scs.2015.09.011, 2016.

*Liu, J., Doan, C. D., Liong, S.-Y., Sanders, R., Dao, A. T., and Fewtrell, T.: Regional frequency analysis of extreme rainfall events in Jakarta, Nat. Hazards, 75, 1075–1104, https://doi.org/10.1007/s11069-014-1363-5, 2015.

Lizardo, J., Prabowo, D. H., Furinto, D. A., and Budiastuti, D. D.: Market Attractiveness And Collaboration Strategies In Improving The Business Performance Of The Digital Out Of Home Media Industry In Indonesia, Int. J. Sci. Technol. Res., 8, 8, 2019.

Lugina, F. P., Riawan, E., and Renggono, F.: The effect of moving rainstorm in increasing river discharge in Ciliwung basin, case study: 15-16 January 2013 flood events, in: AIP Conference Proceedings 1987, INTERNATIONAL SYMPOSIUM ON EARTH HAZARD AND DISASTER MITIGATION (ISEDM) 2017: The 7th Annual Symposium on Earthquake and Related Geohazard Research for Disaster Risk Reduction, Bandung, Indonesia, https://doi.org/10.1063/1.5047338, 2018.

*Mahanani, W. and Chotib: The influence of collective action, community empowerment, and shared vision to the community
capacity in urban water resource conservation, IOP Conf. Ser. Earth Environ. Sci., 200, doi:10.1088/1755-1315/200/1/012040, 2018.

Mangkuto, R. A., Rachman, A. P., Aulia, A. G., Asri, A. D., and Rohmah, M.: Assessment of pitch floodlighting and glare condition in the Main Stadium of Gelora Bung Karno, Indonesia, Measurement, 117, 186–199, https://doi.org/10.1016/j.measurement.2017.12.016, 2018.

*Mantasa Salve Prastica, R.: The analysis of Ancol polder system as flood prevention infrastructure in Jakarta, MATEC Web Conf., 195, https://doi.org/10.1051/matecconf/201819505008, 2018.

*Mardjono, A. and Setiawan, F.: The Advantages of Dry Dam as Flood Control in the Urban Area, Twenty-Sixth International Congress on Large Dams, Vienna, Austria, 3, https://doi.org/10.1201/9780429465086, 2018.

*Mardjono, A., Tri Juwon, P., Montarcih Limantara, L., and Suhartan, E.: Effectivity of kiwi and sukamahi dam on jakarta
flood control, Int. J. Eng. Technol., 7, 134–137, https://doi.org/10.14419/ijet.v7i3.29.18541, 2018.

*Marfai, M. A., Sekaranom, A. B., and Ward, P.: Community responses and adaptation strategies toward flood hazard in Jakarta, Indonesia, Nat. Hazards, 75, 1127–1144, https://doi.org/10.1007/s11069-014-1365-3, 2015.

Marko, K., Zulkarnain, F., and Kusratmoko, E.: Coupling of Markov chains and cellular automata spatial models to predict land cover changes (case study: upper Ci Leungsi catchment area), IOP Conf. Ser. Earth Environ. Sci., 47, 012032,
https://doi.org/10.1088/1755-1315/47/1/012032, 2016.

*Marko, K., Kusratmoko, E., Parlindungan Tambunan, M., and Pahlevi, R.: A Spatial Approach in Assessing Flood Losses in Floodplain Area of Pesanggrahan River (Case Study on Ulujami and Cipulir Urban Villages, South Jakarta), IOP Conf. Ser. Earth Environ. Sci., 338, 012030, https://doi.org/10.1088/1755-1315/338/1/012030, 2019a.

Marko, K., Kusratmoko, E., Tambunan, M. P., and Pahlevi, R.: A Spatial Approach in Assessing Flood Losses in Floodplain
Area of Pesanggrahan River (Case Study on Ulujami and Cipulir Urban Villages, South Jakarta), IOP Conf. Ser. Earth Environ. Sci., 338, 012030, https://doi.org/10.1088/1755-1315/338/1/012030, 2019b.

*Mathewson, D. W.: Historic Institutionalism and Urban Morphology in Jakarta: Moving Towards Building Flood Resiliency into the Formal Planning and Development System, J. Reg. City Plan., 29, 188–209, https://doi.org/10.5614/jrcp.2018.29.3.2, 2018.

Mbarep, D. P. P. and Herdiansyah, H.: Ecological function of green open space as water infiltration: study in kalijodo green open space, north jakarta, J. Phys. Conf. Ser., 1381, 012049, https://doi.org/10.1088/1742-6596/1381/1/012049, 2019.

*Mishra, B. K., Rafiei Emam, A., Masago, Y., Kumar, P., Regmi, R. K., and Fukushi, K.: Assessment of future flood inundations under climate and land use change scenarios in the Ciliwung River Basin, Jakarta: Assessment of future flood

inundations under climate and land use change scenarios in the Ciliwung River Basin, Jakarta, J. Flood Risk Manag., 11, S1105–S1115, https://doi.org/10.1111/jfr3.12311, 2018.

*Moe, I. R., Kure, S., Januriyadi, N. F., Kazama, S., Udo, K., and Koshimura, S.: Development of a Rainfall Runoff and Flood Inundation Model for Jakarta, Indonesia, and Its Sensitivity Analysis of Datasets to Flood Inundation, in: World Environmental and Water Resources Congress 2017, World Environmental and Water Resources Congress 2017, Sacramento, California, 104–116, https://doi.org/10.1061/9780784480601.010, 2017.

*Mohajit: Mathematical Modelling of Injection Wells for Flooding Prevention in Jakarta, Procedia Eng., 125, 207–212, https://doi.org/10.1016/j.proeng.2015.11.030, 2015.

Mulyani Sunarharum, T., Sloan, M., and Susilawati, C.: Re-framing planning decision-making: increasing flood resilience in Jakarta, Int. J. Disaster Resil. Built Environ., 5, 230–242, https://doi.org/10.1108/IJDRBE-02-2014-0015, 2014.

Murdiana, A. W., Soesilo, T., and Bismo, S.: Feasibility study of rainwater conservation and harvesting for industrial community in North Jakarta, IOP Conf. Ser. Earth Environ. Sci., 311, 012056, https://doi.org/10.1088/1755-1315/311/1/012056, 2019.

Nataadmadja, A. D., Prahara, E., and Setyandito, O.: The Effect of Moisture on Hot Mix Asphalt – Case of Indonesian Aggregates, Int. J. Eng. Adv. Technol., 8, 7, 2019.

Neise, T. and Revilla Diez, J.: Firms' contribution to flood risk reduction – scenario-based experiments from Jakarta and Semarang, Indonesia, Procedia Eng., 212, 567–574, https://doi.org/10.1016/j.proeng.2018.01.073, 2018a.

*Neise, T. and Revilla Diez, J.: Überschwemmungen und Regionalentwicklung, Geogr. Rundsch., April, 6, 2018b.

*Neise, T. and Revilla Diez, J.: Adapt, move or surrender? Manufacturing firms' routines and dynamic capabilities on flood risk reduction in coastal cities of Indonesia, Int. J. Disaster Risk Reduct., 33, 332–342, https://doi.org/10.1016/j.ijdrr.2018.10.018, 2019.

Neise, T., Revilla Diez, J., and Garschagen, M.: Firms as drivers of integrative adaptive regional development in the context of environmental hazards in developing countries and emerging economies – A conceptual framework, Environ. Plan. C Polit. Space, 36, 1522–1541, https://doi.org/10.1177/2399654418771079, 2018.

Neise, T., Sambodo, M. T., and Revilla Diez, J.: Are Micro-, Small- and Medium-Sized Enterprises Willing to Contribute to Collective Flood Risk Reduction? Scenario-Based Field Experiments from Jakarta and Semarang, Indonesia, Organ. Environ., https://doi.org/10.1177/1086026619875435, 2019.

*Neolaka, A.: Flood disaster risk in Jakarta, Indonesia, in: Flood Recovery. Innovation and Response III, vol. 159, WIT Press, 107–118, 2012.

*Neolaka, A.: Stakeholder participation in flood control of Ciliwung river, Jakarta, Indonesia, in: Water Resources Management VII, vol. 171, WIT Press, New Forest, UK, 275–285, 2013.

Nilan, P.: Discourses of Non-Formal Pedagogy in Two Youth-Oriented Indonesian Environmental NGOs, Asian Soc. Sci., 11, 162–173, https://doi.org/10.5539/ass.v11n21p162, 2015.

Noviadriana, D., Andawayanti, U., Juwono, P. T., and Sisinggih, D.: Indicators of Index for Polder Services use Partial Least Square and Personal Component Analysis Method, IOP Conf. Ser. Earth Environ. Sci., 437, 012028, https://doi.org/10.1088/1755-1315/437/1/012028, 2020.

*Noviandi, T. U. Z., Kaswanto, R. L., and Arifin, H. S.: Riparian landscape management in the midstream of Ciliwung River as supporting Water Sensitive Cities program with priority of productive landscape, IOP Conf. Ser. Earth Environ. Sci., 91, https://doi.org/10.1088/1755-1315/91/1/012033, 2017.

    *Nugroho, J., Soekarno, I., and Harlan, D.: Model of Ciliwung River Flood Diversion Tunnel Using HEC-RAS Software, MATEC Web Conf., 147, 03001, https://doi.org/10.1051/matecconf/201814703001, 2018.

*Nurhidayah, L. and McIlgorm, A.: Coastal adaptation laws and the social justice of policies to address sea level rise: An Indonesian insight, Ocean Coast. Manag., 171, 11–18, https://doi.org/10.1016/j.ocecoaman.2019.01.011, 2019.

    *Nuryanto, D. E., Pawitan, H., Hidayat, R., and Aldrian, E.: Propagation of convective complex systems triggering potential flooding rainfall of Greater Jakarta using satellite data, IOP Conf. Ser. Earth Environ. Sci., 54, 012028, https://doi.org/10.1088/1755-1315/54/1/012028, 2017.

Nuryanto, D. E., Pawitan, H., Hidayat, R., and Aldrian, E.: Contribution of land use changes to meteorological parameters in Greater Jakarta: Case 17 January 2014, IOP Conf. Ser. Earth Environ. Sci., 149, https://doi.org/10.1088/1755-1315/149/1/012028, 2018a.

    Nuryanto, D. E., Hidayat, R., Pawitan, H., and Aldrian, E.: The evolution of Mesoscale Convective System (MCS) around the Greater Jakarta area on 9 February 2015 using MTSAT Satellite, 10, 2018b.

*Nuswantoro, R., Diermanse, F., and Molkenthin, F.: Probabilistic flood hazard maps for Jakarta derived from a stochastic rain-storm generator: Probabilistic flood hazard maps for Jakarta, J. Flood Risk Manag., 9, 105–124, https://doi.org/10.1111/jfr3.12114, 2016.

    *Octavianti, T. and Charles, K.: Disaster Capitalism? Examining the Politicisation of Land Subsidence Crisis in Pushing Jakarta's Seawall Megaproject, Water Altern., 11, 394–420, 2018.

*Octavianti, T. and Charles, K.: The evolution of Jakarta's flood policy over the past 400 years: The lock-in of infrastructural solutions, Environ. Plan. C Polit. Space, 37, 1102–1125, https://doi.org/10.1177/2399654418813578, 2019.

    *Ogie, R., Holderness, T., Dunbar, M., and Turpin, E.: Spatio-topological network analysis of hydrological infrastructure as a decision support tool for flood mitigation in coastal mega-cities, Environ. Plan. B Urban Anal. City Sci., 44, 718–739, https://doi.org/10.1177/0265813516637608, 2016a.

Ogie, R. I. and Forehead, H.: Investigating the accuracy of georeferenced social media data for flood mapping: The PetaJakarta.org case study, in: 2017 4th International Conference on Information and Communication Technologies for Disaster Management (ICT-DM), 2017 4th International Conference on Information and Communication Technologies for Disaster Management (ICT-DM), 1–6, https://doi.org/10.1109/ICT-DM.2017.8275672, 2017.

    *Ogie, R. I., Holderness, T., Dunn, S., and Turpin, E.: Vulnerability analysis of hydrological infrastructure to flooding in
coastal cities - a graph theory approach, in: Transforming the Future of Infrastructure through Smarter Information: Proceedings of the International Conference on Smart Infrastructure and Construction, 27–29 June 2016, 8, 2016b.

    Ogie, R. I., Dunn, S., Holderness, T., and Turpin, E.: Assessing the vulnerability of pumping stations to trash blockage in coastal mega-cities of developing nations, Sustain. Cities Soc., 28, 53–66, https://doi.org/10.1016/j.scs.2016.08.022, 2017a.

    Ogie, R. I., Shukla, N., Sedlar, F., and Holderness, T.: Optimal placement of water-level sensors to facilitate data-driven
management of hydrological infrastructure assets in coastal mega-cities of developing nations, Sustain. Cities Soc., 35, 385–395, https://doi.org/10.1016/j.scs.2017.08.019, 2017b.

Ogie, R. I., Holderness, T., Dunn, S., and Turpin, E.: Assessing the vulnerability of hydrological infrastructure to flood damage in coastal cities of developing nations, Comput. Environ. Urban Syst., 68, 97–109, https://doi.org/10.1016/j.compenvurbsys.2017.11.004, 2018a.

Ogie, R. I., Forehead, H., Clarke, R. J., and Perez, P.: Participation Patterns and Reliability of Human Sensing in Crowd-Sourced Disaster Management, Inf. Syst. Front., 20, 713–728, https://doi.org/10.1007/s10796-017-9790-y, 2018b.

Ogie, R. I., Clarke, R. J., Forehead, H., and Perez, P.: Crowdsourced social media data for disaster management: Lessons from the PetaJakarta.org project, Comput. Environ. Urban Syst., 73, 108–117, https://doi.org/10.1016/j.compenvurbsys.2018.09.002, 2019.

Oktafiani, Putri. M., Jariyah, A., Fitri, Sari. R., and Takako, Hashimoto.: Social media analysis for Indonesian language: case study flood in Jakarta, in: 2012 International Conference on Advanced Computer Science and Information Systems (ICACSIS), 2012 International Conference on Advanced Computer Science and Information Systems (ICACSIS), 161–166, 2012.

*Otsuka, S., Trilaksono, N. J., and Yoden, S.: Comparing Simulated Size Distributions of Precipitation Systems at Different
Model Resolution, Sci. Online Lett. Atmosphere, 13, 130–134, https://doi.org/10.2151/sola.2017-024, 2017.

Padawangi, R.: Chapter 13 Climate Change and the North Coast of Jakarta: Environmental Justice and the Social Construction of Space in Urban Poor Communities, in: Research in Urban Sociology, vol. 12, edited by: Holt, W. G., Emerald Group Publishing Limited, 321–339, https://doi.org/10.1108/S1047-0042(2012)0000012016, 2012.

*Padawangi, R. and Douglass, M.: Water, Water Everywhere: Toward Participatory Solutions to Chronic Urban Flooding in
Jakarta, Pac. Aff., 88, 517–550, https://doi.org/10.5509/2015883517, 2015.

*Padawangi, R., Turpin, E., Herlily, Prescott, M. F., Lee, I., and Shepherd, A.: Mapping an alternative community river: The case of the Ciliwung, Sustain. Cities Soc., 20, 147–157, https://doi.org/10.1016/j.scs.2015.09.001, 2016.

*Park, H., Kwon, S., and Hadi, S.: Land Subsidence Survey and Policy Development in Pantai Mutiara, Jakarta Bay, Indonesia, J. Coast. Res., 75, 1447–1451, https://doi.org/10.2112/SI75-300.1, 2016.

Parlindungan Tambunan, M.: Characteristic of rainfall in the flood period in DKI Jakarta in 1996, 2002, and 2007, MATEC Web Conf, 229, https://doi.org/10.1051/matecconf/201822902012, 2018.

Pearson, S., Windupranata, W., Pranowo, S. W., Putri, A., Ma, Y., Vila-Concejo, A., Fernández, E., Méndez, G., Banks, J., Knights, A. M., Firth, L. B., Breen, B. B., Jarvis, R., Aguirre, J. D., Chen, S., Smith, A. N. H., Steinberg, P., Chatzinikolaou, E., and Arvanitidis, C.: Conflicts in some of the World harbours: what needs to happen next?, Marit. Stud., 15, 10,
https://doi.org/10.1186/s40152-016-0049-x, 2016.

Perez, P., Holderness du Chemin, T., Turpin, E., and Clarke, R.: Citizen-Driven Flood Mapping in Jakarta: A Self-Organising Socio-technical System, in: 2015 IEEE International Conference on Self-Adaptive and Self-Organizing Systems Workshops, 2015 IEEE International Conference on Self-Adaptive and Self-Organizing Systems Workshops, 174–178, https://doi.org/10.1109/SASOW.2015.40, 2015.

Permanasari, E.: Reading Political Insinuation in Urban Forms: Saving the Sinking Jakarta Through Giant Sea Wall Project, Geogr. Tech., 14, 56–65, https://doi.org/10.21163/GT_2019. 141.19, 2019.

Phanuwan, C., Takizawa, S., Oguma, K., Katayama, H., Yunika, A., and Ohgaki, S.: Monitoring of human enteric viruses and coliform bacteria in waters after urban flood in Jakarta, Indonesia, Water Sci. Technol. J. Int. Assoc. Water Pollut. Res., 54, 203–210, https://doi.org/10.2166/wst.2006.470, 2006.

Poerbandono, Julian, M. M., and Ward, P. J.: Assessment of the effects of climate and land cover changes on river discharge and sediment yield, and an adaptive spatial planning in the Jakarta region, Nat. Hazards, 73, 507–530, https://doi.org/10.1007/s11069-014-1083-x, 2014.

Pradafitri, W. S. and Moersidik, S. S.: Potential of east flood canal as provider of drinking water ecosystem services for DKI Jakarta, IOP Conf. Ser. Earth Environ. Sci., 306, 012005, https://doi.org/10.1088/1755-1315/306/1/012005, 2019.

Pradafitri, W. S., Moersidik, S. S., and Abdini, C.: East Canal Flood as PDAM water resource DKI Jakarta, E3S Web Conf., 74, 09001, https://doi.org/10.1051/e3sconf/20187409001, 2018.

Pradafitri, W. S., Moersidik, S. S., and Abdini, C.: MBBR technology for water treatment east flood canal as a raw water source of DKI Jakarta, IOP Conf. Ser. Earth Environ. Sci., 311, 012058, https://doi.org/10.1088/1755-1315/311/1/012058, 2019.

*Prasetyo, Y., Yuwono, B. D., and Ramadhanis, Z.: Spatial Analysis of Land Subsidence and Flood Pattern Based on DInSAR Method in Sentinel Sar Imagery and Weighting Method in Geo-Hazard Parameters Combination in North Jakarta Region, IOP Conf. Ser. Earth Environ. Sci., 123, 012009, https://doi.org/10.1088/1755-1315/123/1/012009, 2018.

Prastica, R. M. S., Apriatresnayanto, R., and Marthanty, D. R.: Structural and Green Infrastructure Mitigation Alternatives Prevent Ciliwung River from Water-related Landslide, 9, n.d.

Pratama, M.: Tidal Flood in Pekalongan, Mater. Sci. Eng., 11, 2019.

Pratita Sari, D., Susiloningtyas, D., and Rizqihandari, N.: Drought Adaptation of Water Usage in Regular and Irregular Settlements (Case Study: Jatinegara District, East Jakarta), IOP Conf. Ser. Earth Environ. Sci., 338, 012022, https://doi.org/10.1088/1755-1315/338/1/012022, 2019.

Pravitasari, A. E., Rustiadi, E., Mulya, S. P., Setiawan, Y., Fuadina, L. N., and Murtadho, A.: Identifying the driving forces of 1075     urban expansion and its environmental impact in Jakarta-Bandung mega urban region, IOP Conf. Ser. Earth Environ. Sci., 149, 012044, https://doi.org/10.1088/1755-1315/149/1/012044, 2018.

*Priambodo, I., Tambunan, M. P., and Kusratmoko, E.: Spatial and statistical analysis on the cause of flooding in Northwest Jakarta floodplain (Kapuk and Penjaringan Districts), MATEC Web Conf., 229, https://doi.org/10.1051/matecconf/201822904008, 2018.

Pribadi, D. O., Vollmer, D., and Pauleit, S.: Impact of peri-urban agriculture on runoff and soil erosion in the rapidly developing metropolitan area of Jakarta, Indonesia, Reg. Environ. Change, 18, 2129–2143, https://doi.org/10.1007/s10113-018-1341-7, 2018.

*Purba, F. D., Hunfeld, J. A. M., Fitriana, T. S., Iskandarsyah, A., Sadarjoen, S. S., Busschbach, J. J. V., and Passchier, J.: Living in uncertainty due to floods and pollution: the health status and quality of life of people living on an unhealthy riverbank, 1085     BMC Public Health, 18, 1–11, https://doi.org/10.1186/s12889-018-5706-0, 2018.

Purbiati, T. and Santoso, P.: Consumer acceptance of nine varieties of cut rose flower grown in dry and highland areas of Indonesia, Acta Hortic., 499–504, https://doi.org/10.17660/ActaHortic.2007.755.69, 2007.

Purnomo, R., Pamungkas, M. H., Arrofi, D., and Goni, A.: Flood prediction using integrated sensor based on internet of thing and radio frequency as flood risk reduction, AIP Conf. Proc., 1987, 020070, https://doi.org/10.1063/1.5047355, 2018.

Pusponegoro, A. D.: Terrorism in Indonesia, Prehospital Disaster Med., 18, 100–105, https://doi.org/10.1017/S1049023X00000832, 2003.

Rachello-Dolmen, P. G. and Cleary, D. F. R.: Relating coral species traits to environmental conditions in the Jakarta Bay/Pulau Seribu reef system, Indonesia, Estuar. Coast. Shelf Sci., 73, 816–826, https://doi.org/10.1016/j.ecss.2007.03.017, 2007.

Rachmat, F. B. and Muhammad, H.: Analysis of land use changes to the criticality level of the catchment area in eight 1095 watersheds that flow into Jakarta Bay, Indonesia, E3S Web Conf., 73, 03001, https://doi.org/10.1051/e3sconf/20187303001, 2018.

*Rafiei Emam, A., Mishra, B., Kumar, P., Masago, Y., and Fukushi, K.: Impact Assessment of Climate and Land-Use Changes on Flooding Behavior in the Upper Ciliwung River, Jakarta, Indonesia, Water, 8, 559, https://doi.org/10.3390/w8120559, 2016.

*Rahayu, H. P., Haigh, R., Amaratunga, D., Kombaitan, B., Khoirunnisa, D., and Pradana, V.: A micro scale study of climate 1100 change adaptation and disaster risk reduction in coastal urban strategic planning for the Jakarta, Int. J. Disaster Resil. Built Environ., 11, 15, 2020.

*Rahmayati, Y., Parnell, M., and Himmayani, V.: Understanding community-led resilience: The Jakarta floods experience, Aust. J. Emerg. Manag., 32, 58–66, 2017.

Randi, C., Andromeda, Z. I., Nazhifah, K., Syahputra, R., Mahmuddin, I., and Septyandy, M. R.: Massive earthquake 1105 countermeasures by establish muster point and migration path using network analysis in Matraman District, Jakarta, Indonesia, IOP Conf. Ser. Mater. Sci. Eng., 620, 012121, https://doi.org/10.1088/1757-899X/620/1/012121, 2019.

Randil, C., Siriwardana, C., and Dias, P.: Comparison of Damage Values Used in Different Flood Modelling Studies, in: 2019 Moratuwa Engineering Research Conference (MERCon), 2019 Moratuwa Engineering Research Conference (MERCon), 25–30, https://doi.org/10.1109/MERCon.2019.8818933, 2019.

*Remondi, F., Burlando, P., and Vollmer, D.: Exploring the hydrological impact of increasing urbanisation on a tropical river catchment of the metropolitan Jakarta, Indonesia, Sustain. Cities Soc., 20, 210–221, https://doi.org/10.1016/j.scs.2015.10.001, 2016.

Ristiani, C. R., Rokhmatuloh, and Hernina, R.: Neighbourhood Socio Economic Disadvantage Index's Analysis of the Flood Disasters Area at East Jakarta in 1996 and 2016, IOP Conf. Ser. Earth Environ. Sci., 98, https://doi.org/10.1088/1755-1115 1315/98/1/012007, 2017.

Ristriyani, R., Nur Rachmawati, I., and Afiyanti, Y.: Status disclosure and the acceptance of women living with HIV, Enfermeria Clin., 28 Suppl 1, 195–198, https://doi.org/10.1016/S1130-8621(18)30066-4, 2018.

*Riyando Moe, I., Kure, S., Fajar Januriyadi, N., Farid, M., Udo, K., Kazama, S., and Koshimura, S.: Future projection of flood inundation considering land-use changes and land subsidence in Jakarta, Indonesia, Hydrol. Res. Lett., 11, 99–105, 1120 https://doi.org/10.3178/hrl.11.99, 2017.

Riyanto, I., Margatama, L., Arief, R., Sudiana, D., and Sudibyo, H.: Pesanggrahan River Watershed Flood Potential Mapping in South and West Jakarta with LiDAR Data Segmentation, J. Phys. Conf. Ser., 1201, 012028, https://doi.org/10.1088/1742-6596/1201/1/012028, 2019.

Rizqi, Z. U., Putri, B. A. D., Sabit, M. I., and Salsabila, S. E.: Establishment of Magnetic Levitation for Flood Prevention in Jakarta with Project Management Approach, 10, 2019.

Rofiq Ginanjar, M. and Sandy Putra, S.: Sediment trapping analysis of flood control reservoirs in Upstream Ciliwung River using SWAT Model, IOP Conf. Ser. Earth Environ. Sci., 71, https://doi.org/10.1088/1755-1315/71/1/012014, 2017.

*Rojali, A., Budiaji, A. S., Pribadi, Y. S., Fatria, D., and Hadi, T. W.: A preliminary comparison of hydrodynamic approaches for flood inundation modeling of urban areas in Jakarta Ciliwung river basin, INTERNATIONAL SYMPOSIUM ON EARTH HAZARD AND DISASTER MITIGATION (ISEDM) 2016: The 6th Annual Symposium on Earthquake and Related Geohazard Research for Disaster Risk Reduction, Bandung, Indonesia, https://doi.org/10.1063/1.4987099, 2017.

Rositasari, R., Puspitasari, R., and Purbonegoro, T.: The use of ammonia-elphidium (A-E) index in Jakarta Bay and Semarang coastal waters, Indonesia, 9, n.d.

Rosyidin, W. F., Dahlia, S., Zahro, A. A., Putra, A. R., Katami, M., and Najiyullah, M.: Identify of Multi-Hazard on Muhammadiyah Education Area by VISUS Method in Jakarta, in: IOP Conf. Series: Earth and Environmental Science, ICHMGEP, https://doi.org/10.1088/1755-1315/271/1/012015, 2019.

Rumanta, M.: The potential of Rhizophora mucronata and Sonneratia caseolaris for phytoremediation of lead pollution in Muara Angke, North Jakarta, Indonesia, Biodiversitas J. Biol. Divers., 20, https://doi.org/10.13057/biodiv/d200808, 2019.

*Rusdiansyah, A., Tang, Y., He, Z., Li, L., Ye, Y., and Yahya Surya, M.: The impacts of the large-scale hydraulic structures on tidal dynamics in open-type bay: numerical study in Jakarta Bay, Ocean Dyn., 68, 1141–1154, https://doi.org/10.1007/s10236-018-1183-3, 2018.

Saddhono, K. and Sulaksono, D.: Indoglish as adaptation of english to Indonesian: change of society in big cities of Indonesia, IOP Conf. Ser. Earth Environ. Sci., 126, 012092, https://doi.org/10.1088/1755-1315/126/1/012092, 2018.

Sagala, S., Syahbid, M., and Wibisono, H.: The role of leaders in risk governance in Jakarta, in: Jakarta, edited by: Hellman, J., Thynell, M., and van Voorst, R., Routledge, London, 2018.

*Salim, W., Bettinger, K., and Fisher, M.: Maladaptation on the Waterfront: Jakarta's Growth Coalition and the Great Garuda, Environ. Urban. ASIA, 10, 63–80, https://doi.org/10.1177/0975425318821809, 2019.

Sanditya Hardaya, I. B. N., Dhini, A., and Surjandari, I.: Application of text mining for classification of community complaints and proposals, in: 2017 3rd International Conference on Science in Information Technology (ICSITech), 2017 3rd International Conference on Science in Information Technology (ICSITech), Bandung, Indonesia, 144–149, https://doi.org/10.1109/ICSITech.2017.8257100, 2017.

Sardjono, W. and Perdana, W. G.: The Application of Artificial Neural Network for Flood Systems Mitigation at Jakarta City, in: 2019 International Conference on Information Management and Technology (ICIMTech), 2019 International Conference on Information Management and Technology (ICIMTech), 137–140, https://doi.org/10.1109/ICIMTech.2019.8843735, 2019.

*Sari, D. A. P., Madonna, S., and Fitriani, A.: Environmental Health Evaluation for Jatinegara Apartment from the Perception of Kampung Pulo Displaced People, Int. J. Eng. Technol., 7, 224–228, https://doi.org/10.31227/osf.io/a58ht, 2018a.

Sari, D. A. P., Sugiana, A., Ramadhonah, R. Y., Innaqa, S., and Rahim, R.: Kampung Pulo Environmental Planning Observed from Biophysical Aspects as Adaptation of Flood in Jakarta, Int. J. Eng. Technol., 7, 82–87, https://doi.org/10.14419/ijet.v7i2.3.12621, 2018b.

Sari, V. F. M., Sutjiningsih, D., and Anggraheni, E.: Effectiveness of Muara Angke Polder System in North Jakarta, Int. J. Innov. Technol. Explor. Eng., 8, 5, 2019.

*Saridewi, T. R. and Fauzi, A.: A market-based mechanism as an alternative solution for watershed management: a case study of the Ciliwung Watershed, Indonesia, Int. J. Glob. Environ. Issues, 18, 171, https://doi.org/10.1504/IJGENVI.2019.10023932, 2019.

Septiana, M. N. and Gayatri, D.: The impact of discomfort: Physical and psychological to social interaction in diabetic ulcer patients in Jakarta - Indonesia, Enfermeria Clin., 29 Suppl 2, 407–412, https://doi.org/10.1016/j.enfcli.2019.04.059, 2019.

Setyandito, O., Syafalni, S., Wijayanti, Y., C., D., and Satrio, S.: Groundwater Quality Assessment And Recharge Well Design Of Cengkareng Area, West Jakarta, Int. J. Appl. Eng. Res., 2020.

Shaad, K., Ninsalam, Y., Padawangi, R., and Burlando, P.: Towards high resolution and cost-effective terrain mapping for
urban hydrodynamic modelling in densely settled river-corridors, Sustain. Cities Soc., 20, 168–179, https://doi.org/10.1016/j.scs.2015.09.005, 2016.

*Shatkin, G.: Futures of Crisis, Futures of Urban Political Theory: Flooding in Asian Coastal Megacities: Flood Risk And Littoral Conurbations, Int. J. Urban Reg. Res., 43, 207–226, https://doi.org/10.1111/1468-2427.12758, 2019.

*Sheppard, E.: Globalizing capitalism's raggedy fringes: thinking through Jakarta, Area Dev. Policy, 4, 1–27,
https://doi.org/10.1080/23792949.2018.1523682, 2019.

*Shokhrukh-Mirzo Jalilov, Mohamed Kefi, Pankaj Kumar, Yoshifumi Masago, and Binaya Mishra: Sustainable Urban Water Management: Application for Integrated Assessment in Southeast Asia, Sustainability, 10, 122, https://doi.org/10.3390/su10010122, 2018.

*Sholichin, M., Prayogo, T. B., and Bisri, M.: Using HEC-RAS for analysis of flood characteristic in Ciliwung River,
Indonesia, IOP Conf. Ser. Earth Environ. Sci., 344, 012011, https://doi.org/10.1088/1755-1315/344/1/012011, 2019.

Silver, C.: Waterfront Jakarta: The battle for the future of the metropolis, in: Jakarta, edited by: Hellman, J., Thynell, M., and van Voorst, R., Routledge, London, 2018.

*Simanjuntak, I., Frantzeskaki, N., Enserink, B., and Ravesteijn, W.: Evaluating Jakarta's flood defence governance: the impact of political and institutional reforms, Water Policy, 14, 561–580, https://doi.org/10.2166/wp.2012.119, 2012.

*Simarmata, H. A.: Phenomenology in Adaptation Planning, Springer Singapore, Singapore, https://doi.org/10.1007/978-981-10-5496-9, 2018.

Simone, A.: What You See is Not Always What You Know, South East Asia Res., 23, 227–244, https://doi.org/10.5367/sear.2015.0258, 2015.

Simone, A.: Urbanity and Generic Blackness, Theory Cult. Soc., 33, 183–203, https://doi.org/10.1177/0263276416636203,
2016.

Siswanto, Oldenborgh, G. J. van, Schrier, G. van der, Lenderink, G., and Hurk, B. van den: Trends in High-Daily Precipitation Events in Jakarta and the Flooding of January 2014, Bull. Am. Meteorol. Soc., 96, S131–S135, https://doi.org/10.1175/BAMS-D-15-00128.1, 2015.

Siswanto, van der Schrier, G., Jan van Oldenborgh, G., van den Hurk, B., Aldrian, E., Swarinoto, Y., Sulistya, W., and Eka
Sakya, A.: A very unusual precipitation event associated with the 2015 floods in Jakarta: an analysis of the meteorological factors, Weather Clim. Extrem., 16, 23–28, https://doi.org/10.1016/j.wace.2017.03.003, 2017.

Sitinjak, E., Meidityawati, B., Ichwan, R., Onggosandojo, N., and Aryani, P.: Enhancing Urban Resilience through Technology and Social Media: Case Study of Urban Jakarta, Procedia Eng., 212, 222–229, https://doi.org/10.1016/j.proeng.2018.01.029, 2018.

*Soemabrata, J.: RISK MAPPING STUDIES OF HYDRO-METEOROLOGICAL HAZARD IN DEPOK MIDDLE CITY, Int. J. GEOMATE, 14, https://doi.org/10.21660/2018.44.3730, 2018.

Sriratnasari, S. R., Sfenrianto, Fajar, A. N., Nurcahyo, A., and Albert: Drone Utilization for Jakarta as a Smart City, J. Phys. Conf. Ser., 1179, 012117, https://doi.org/10.1088/1742-6596/1179/1/012117, 2019.

*Su, H.-T., Cheung, S. H., and Lo, E. Y.-M.: Multi-objective optimal design for flood risk management with resilience
objectives, Stoch. Environ. Res. Risk Assess., 32, 1147–1162, https://doi.org/10.1007/s00477-017-1508-7, 2018.

*Sugar, L., Kennedy, C., and Hoornweg, D.: Synergies between climate change adaptation and mitigation in development: Case studies of Amman, Jakarta, and Dar es Salaam, Int. J. Clim. Change Strateg. Manag., 5, 95–111, https://doi.org/10.1108/17568691311299381, 2013.

*Sujono, J.: Hydrological Analysis of the Situ Gintung Dam Failure, J. Disaster Res., 7, 590–594,
https://doi.org/10.20965/jdr.2012.p0590, 2012.

Sukmana, H. T., Ichsani, Y., and Putra, S. J.: Implementation of Server Consolidation Method on a Data Center by using Virtualization Technique: A Case Study, in: 2016 International Conference on Informatics and Computing (ICIC), 2016 International Conference on Informatics and Computing (ICIC), 277–282, https://doi.org/10.1109/IAC.2016.7905729, 2016.

Sulistiowati, N. M. D., Keliat, B. A., Wardani, I. Y., Aldam, S. F. S., Triana, R., and Florensa, M. V. A.: Comprehending
Mental Health in Indonesian's Adolescents through Mental, Emotional, and Social Well-Being, Compr. Child Adolesc. Nurs., 42, 277–283, https://doi.org/10.1080/24694193.2019.1594460, 2019.

Sumargo, B. and Novalia, T.: Structural Equation Modelling for Determining Subjective Well-Being Factors of the Poor Children in Bad Environment, Procedia Comput. Sci., 135, 113–119, https://doi.org/10.1016/j.procs.2018.08.156, 2018.

Sunaryo, L.: Institutionalizing complexity and inclusivity for developing social entrepreneurship, J. Institutional Res. South
East Asia, 13, 55–67, 2015.

*Suprayogi, H., Rudyanto, A., Bachtiar, H., and Limantara, L. M.: Critical-phase sea dike construction of NCICD program in Jakarta as national capital city, IOP Conf. Ser. Earth Environ. Sci., 162, 012020, https://doi.org/10.1088/1755-1315/162/1/012020, 2018.

Suryo, P. and Murachman, B.: Development of Non Petroleum Base Chemicals for Improving Oil Recovery in Indonesia, SPE
Asia Pacific Oil and Gas Conference and Exhibition, https://doi.org/10.2118/68768-MS, 2001.

Susanti, H., Hamid, A. Y. S., Mulyono, S., Putri, A. F., and Chandra, Y. A.: Expectations of survivors towards disaster nurses in Indonesia: A qualitative study, Int. J. Nurs. Sci., 6, 392–398, https://doi.org/10.1016/j.ijnss.2019.09.001, 2019.

*Susilo, A. J., Sumarli, I., Sentosa, G. S., Prihatiningasih, A., and Wongkar, E.: Effect of Compaction to Increase the Critical Height of a Slope without any Support, Mater. Sci. Eng., 10, 2019.

Susilowardhani, E. M., Djuhardi, L., and Yulianti, I.: Risk Communication in Reducing Flood Risk in Jakarta, IOP Conf. Ser. Earth Environ. Sci., 145, 012085, https://doi.org/10.1088/1755-1315/145/1/012085, 2018.

*Sutrisno, D.: Modelling the projection of climate change impact on shoreline retreat: remote sensing approach, in: 32nd Asian Conference on Remote Sensing, 32nd Asian Conference on Remote Sensing, Taipei, Taiwan, 74–79, 2011.

Suwartha, N. and Priadi, C.: Modeling Surface Water Quality of UI Recharge Pond using Numerical Method, Int. J. Technol., 1235    4, 136–146, https://doi.org/10.14716/ijtech.v4i2.109, 2013.

Suwarti, R., Moersidik, S. S., and Soesilo, T. E. B.: Environmental & Economic Valuation of Raw Water Resource of East Flood Canal DKI Jakarta, IOP Conf. Ser. Earth Environ. Sci., 306, 012012, https://doi.org/10.1088/1755-1315/306/1/012012, 2019.

*Syafalni, S., Setyandito, O., Lubis, F. R., and Wijayanti, Y.: Frequency Analysis of Design-Flood Discharge Using Gumbel 1240    Distribution at Katulampa Weir, Ciliwung River, Int. J. Appl. Eng. Res., 10, 9935–9946, 2015.

*Takagi, H., Mikami, T., Fujii, D., Esteban, M., and Kurobe, S.: Mangrove forest against dyke-break-induced tsunami on rapidly subsidingcoasts, Nat. Hazards Earth Syst. Sci., 16, 1629–1638, https://doi.org/10.5194/nhess-16-1629-2016, 2016a.

*Takagi, H., Esteban, M., Mikami, T., and Fujii, D.: Projection of coastal floods in 2050 Jakarta, Urban Clim., 17, 135–145, https://doi.org/10.1016/j.uclim.2016.05.003, 2016b.

*Takagi, H., Fujii, D., Esteban, M., and Yi, X.: Effectiveness and Limitation of Coastal Dykes in Jakarta: The Need for Prioritizing Actions against Land Subsidence, Sustainability, 9, 619, https://doi.org/10.3390/su9040619, 2017.

Tambunan, M. P.: Flooding area in the Jakarta province on February 2 to 4 2007: 28th Asian Conference on Remote Sensing 2007, ACRS 2007, 28th Asian Conf. Remote Sens. 2007 ACRS 2007, 1428–1443, 2007.

*Tambunan, M. P.: The pattern of spatial flood disaster region in DKI Jakarta, IOP Conf. Ser. Earth Environ. Sci., 56, 012014, 1250    https://doi.org/10.1088/1755-1315/56/1/012014, 2017.

*Tambunan, M. P.: Characteristic of rainfall in the flood period in DKI Jakarta in 1996, 2002, and 2007, MATEC Web Conf, 229, doi:10.1051/matecconf/201822902012, 2018.

Taniguchi, M.: What are the Subsurface Environmental Problems?, in: Groundwater and Subsurface Environments: Human Impacts in Asian Coastal Cities, edited by: Taniguchi, M., Springer Japan, Tokyo, 3–18, https://doi.org/10.1007/978-4-431-1255    53904-9_1, 2011.

Tanuwidjaja, G. and Chang, B. G.: Green Infrastructure Concept for JABODETABEKJUR Metropolitan Area, IOP Conf. Ser. Earth Environ. Sci., 79, 012024, https://doi.org/10.1088/1755-1315/79/1/012024, 2017.

Tay, J.: The search for an Asian Idol: The performance of regional identity in reality television, Int. J. Cult. Stud., 14, 323–338, https://doi.org/10.1177/1367877910391870, 2011.

Texier-Teixeira, P. and Edelblutte, E.: Jakarta: Mumbai—Two Megacities Facing Floods Engaged in a Marginalization Process of Slum Areas, in: Identifying Emerging Issues in Disaster Risk Reduction, Migration, Climate Change and Sustainable Development: Shaping Debates and Policies, edited by: Sudmeier-Rieux, K., Fernández, M., Penna, I. M., Jaboyedoff, M., and Gaillard, J. C., Springer International Publishing, Cham, 81–99, https://doi.org/10.1007/978-3-319-33880-4_6, 2017.

Theresia, C., Gabe, R. T., and Adianto, J.: Housing preferences and strategies of Javanese migrants in Jakarta, IOP Conf. Ser. Mater. Sci. Eng., 523, 012054, https://doi.org/10.1088/1757-899X/523/1/012054, 2019.

Tiyas, N. E. W. and Sutjiningsih, D.: The Influence of Land Use Change and Spatial Discretization of Middle - Lower Ciliwung Sub-Watershed on Flood Hydrograph at Manggarai Weir : a Preliminary Study, Int. J. Eng. Technol., 7, 497–506, https://doi.org/10.14419/ijet.v7i3.30.18418, 2018.

Trilaksono, N. J.: Numerical Studies of Heavy Precipitation over West Java in January–February 2007, 2012.

Trilaksono, N. J., Otsuka, S., Yoden, S., Saito, K., and Hayashi, S.: Dependence of Model-Simulated Heavy Rainfall on the Horizontal Resolution during the Jakarta Flood Event in January-February 2007, Sola, 7, 193–196, https://doi.org/10.2151/sola.2011-049, 2011.

Trilaksono, N. J., Otsuka, S., and Yoden, S.: A Time-Lagged Ensemble Simulation on the Modulation of Precipitation over West Java in January–February 2007, Mon. Weather Rev., 140, 601–616, https://doi.org/10.1175/MWR-D-11-00094.1, 2012.

Triyana, H. J.: Indonesian compliance and its effective implementation of international norms on disaster response, in: Humanitarian Action, edited by: Zwitter, A., Lamont, C. K., Heintze, H.-J., and Herman, J., Cambridge University Press, Cambridge, 330–348, https://doi.org/10.1017/CBO9781107282100.021, 2014.

van Voorst, R.: Applying the risk society thesis within the context of flood risk and poverty in Jakarta, Indonesia, Health Risk Soc., 17, 246–262, https://doi.org/10.1080/13698575.2015.1071785, 2015.

*van Voorst, R.: The Right to Aid: Perceptions and Practices of Justice in a Flood-Hazard Context in Jakarta, Indonesia, Asia Pac. J. Anthropol., 15, 339–356, https://doi.org/10.1080/14442213.2014.916340, 2014.

*van Voorst, R.: Risk-handling styles in a context of flooding and uncertainty in Jakarta, Indonesia: An analytical framework to analyse heterogenous risk-behaviour, Disaster Prev. Manag. Int. J., 24, 484–505, https://doi.org/10.1108/DPM-04-2014-0065, 2015.

*van Voorst, R.: Formal and informal flood governance in Jakarta, Indonesia, Habitat Int., 52, 5–10, https://doi.org/10.1016/j.habitatint.2015.08.023, 2016.

van Voorst, R.: Natural Hazards, Risk and Vulnerability: Floods and slum life in Indonesia, Routledge, London, 178 pp., https://doi.org/10.4324/9781315716411, 2016.

*van Voorst, R. and Hellman, J.: One Risk Replaces Another, Asian J. Soc. Sci., 43, 786–810, https://doi.org/10.1163/15685314-04306007, 2015.

*Varrani, A. and Nones, M.: Vulnerability, impacts and assessment of climate change on Jakarta and Venice, Int. J. River Basin Manag., 16, 439–447, https://doi.org/10.1080/15715124.2017.1387125, 2018.

Vollaard, A. M., Ali, S., van Asten, H. A. G. H., Widjaja, S., Visser, L. G., Surjadi, C., and van Dissel, J. T.: Risk factors for typhoid and paratyphoid fever in Jakarta, Indonesia, JAMA, 291, 2607–2615, https://doi.org/10.1001/jama.291.21.2607, 2004.

Vollmer, D. and Grêt-Regamey, A.: Rivers as municipal infrastructure: Demand for environmental services in informal settlements along an Indonesian river, Glob. Environ. Change, 23, 1542–1555, https://doi.org/10.1016/j.gloenvcha.2013.10.001, 2013.

*Vollmer, D., Costa, D., Lin, E. S., Ninsalam, Y., Shaad, K., Prescott, M. F., Gurusamy, S., Remondi, F., Padawangi, R., Burlando, P., Girot, C., Grêt-Regamey, A., and Rekittke, J.: Changing the Course of Rivers in an Asian City: Linking Landscapes to Human Benefits through Iterative Modeling and Design, JAWRA J. Am. Water Resour. Assoc., 51, 672–688, https://doi.org/10.1111/1752-1688.12316, 2015.

*Vollmer, D., Pribadi, D. O., Remondi, F., Rustiadi, E., and Grêt-Regamey, A.: Prioritizing ecosystem services in rapidly urbanizing river basins: A spatial multi-criteria analytic approach, Sustain. Cities Soc., 20, 237–252, https://doi.org/10.1016/j.scs.2015.10.004, 2016.

*Wade, M.: Hyper-planning Jakarta: The Great Garuda and planning the global spectacle, Singap. J. Trop. Geogr., 40, 158–172, https://doi.org/10.1111/sjtg.12262, 2019.

*Wahab, R. and Tiong, R.: Multi-variate residential flood loss estimation model for Jakarta: an approach based on a combination of statistical techniques, Nat. Hazards, 86, 779–804, https://doi.org/10.1007/s11069-016-2716-z, 2017.

Wang, D., Xia, H., Liu, Z., and Yang, Q.: Study of the Mechanism of Polymer Solution With Visco-Elastic Behavior Increasing Microscopic Oil Displacement Efficiency and the Forming of Steady Oil Thread Flow Channels, SPE Asia Pacific Oil and Gas Conference and Exhibition, https://doi.org/10.2118/68723-MS, 2001.

*Ward, P. J., Marfai, M. A., Poerbandono, and Aldrian, E.: Climate Adaptation in the City of Jakarta, in: Climate Adaptation and Flood Risk in Coastal Cities, Routledge, 285–304, 2011a.

*Ward, P. J., Marfai, M. A., Yulianto, F., Hizbaron, D. R., and Aerts, J. C. J. H.: Coastal inundation and damage exposure estimation: a case study for Jakarta, Nat. Hazards, 56, 899–916, https://doi.org/10.1007/s11069-010-9599-1, 2011b.

*Ward, P. J., Pauw, W. P., van Buuren, M. W., and Marfai, M. A.: Governance of flood risk management in a time of climate change: the cases of Jakarta and Rotterdam, Environ. Polit., 22, 518–536, https://doi.org/10.1080/09644016.2012.683155, 2013.

*Wicaksono, A. and Herdiansyah, H.: The impact analysis of flood disaster in DKI jakarta: prevention and control perspective, J. Phys. Conf. Ser., 7, https://doi.org/10.1088/1742-6596/1339/1/012092, 2019.

Widjaja, I. A.: Price, E-service quality, cose to customer satisfaction based on e-public service application online technology, Int. J. Appl. Bus. Econ. Res., 15, 507–513, 2017.

Widodo, J., Herlambang, A., Sulaiman, A., Razi, P., Yohandri, Perissin, D., Kuze, H., and Sumantyo, J. T. S.: Land subsidence rate analysis of Jakarta Metropolitan Region based on D-InSAR processing of Sentinel data C-Band frequency, J. Phys. Conf. Ser., 1185, 012004, https://doi.org/10.1088/1742-6596/1185/1/012004, 2019.

Widyahening, I. S., van der Heijden, G. J. M. G., Moy, F. M., van der Graaf, Y., Sastroasmoro, S., and Bulgiba, A.: From west to east; experience with adapting a curriculum in evidence-based medicine, Perspect. Med. Educ., 1, 249–261, https://doi.org/10.1007/s40037-012-0029-9, 2012.

Wigati, R. and Notonegoro, H. A.: Capacity and performance evaluation of the drainage system Jati Pinggir - Petamburan Central Jakarta, IOP Conf. Ser. Mater. Sci. Eng., 673, 012044, https://doi.org/10.1088/1757-899X/673/1/012044, 2019.

*Wihaji, W., Achmad, R., and Nadiroh, N.: Policy evaluation of runoff, erosion and flooding to drainage system in Property Depok City, Indonesia, IOP Conf. Ser. Earth Environ. Sci., 191, 012115, https://doi.org/10.1088/1755-1315/191/1/012115, 2018.

*Wijayanti, P., Zhu, X., Hellegers, P., Budiyono, Y., and van Ierland, E. C.: Estimation of river flood damages in Jakarta, Indonesia, Nat. Hazards, 86, 1059–1079, https://doi.org/10.1007/s11069-016-2730-1, 2017.

Wildsmith, D. V. and Smith, P.: From Bricks to Bytes: Digitizing Green Cities, Int. J. Technol., 4, 269, https://doi.org/10.14716/ijtech.v4i3.114, 2013.

*Wu, P., Arbain, A. A., Mori, S., Hamada, J., Hattori, M., Syamsudin, F., and Yamanaka, M. D.: The Effects of an Active
Phase of the Madden-Julian Oscillation on the Extreme Precipitation Event over Western Java Island in January 2013, Sci. Online Lett. Atmosphere, 9, 79–83, https://doi.org/10.2151/sola.2013-018, 2013.

*van der Wulp, S. A., Dsikowitzky, L., Hesse, K. J., and Schwarzbauer, J.: Master Plan Jakarta, Indonesia: The Giant Seawall and the need for structural treatment of municipal waste water, Mar. Pollut. Bull., 110, 686–693, https://doi.org/10.1016/j.marpolbul.2016.05.048, 2016.

*Wurjanto, A.: Study Of Pump And Retention Basin Requirement For Semarang-Demak Coastal Dike Plan, Central Java, Int. J. GEOMATE, 15, https://doi.org/10.21660/2018.47.68850, 2018.

*Yahya Surya, M., He, Z., Xia, Y., and Li, L.: Impacts of Sea Level Rise and River Discharge on the Hydrodynamics Characteristics of Jakarta Bay (Indonesia), Water, 11, 1384, https://doi.org/10.3390/w11071384, 2019.

Yang, K., Michael, K., Abbas, R., and Holderness, T.: Urban flood modelling using geo-social intelligence, in: 2017 IEEE
International Symposium on Technology and Society (ISTAS), 2017 IEEE International Symposium on Technology and Society (ISTAS), 1–9, https://doi.org/10.1109/ISTAS.2017.8319086, 2017.

*Yayuk Supomo, F., Saleh Pallu, Muh., Arsyad Thaha, Muh., and Tahir Lopa, R.: Determining the Side Channel Area in the Ciliwung Watershed for Decreasing the Hydrograph Flood, IOP Conf. Ser. Earth Environ. Sci., 140, 012038, https://doi.org/10.1088/1755-1315/140/1/012038, 2018.

*Yoga Putra, G. A., Koestoer, R. H., and Lestari, I.: Local resilience towards overcoming floods of local climate change for adaptation: A study of marunda community in north jakarta, IOP Conf. Ser. Earth Environ. Sci., 239, https://doi.org/10.1088/1755-1315/239/1/012043, 2019a.

*Yoga Putra, G. A., Koestoer, R. H., and Lestari, I.: Psycho-social performance towards understanding local adaptation of coastal flood in Cilincing Community, North Jakarta, Indonesia, IOP Conf. Ser. Earth Environ. Sci., 243, 012005,
https://doi.org/10.1088/1755-1315/243/1/012005, 2019b.

Yoo, G., Kim, A. R., and Hadi, S.: A methodology to assess environmental vulnerability in a coastal city: Application to Jakarta, Indonesia, Ocean Coast. Manag., 102, 169–177, https://doi.org/10.1016/j.ocecoaman.2014.09.018, 2014.

*Yuliadi, D., Eriyatno, -, J. Purwanto, M. Y., and Nurjana, I. W.: Socio Economical Impact Analysis and Adaptation Strategy for Coastal Flooding (Case Study on North Jakarta Region), Int. J. Adv. Sci. Eng. Inf. Technol., 6,
https://doi.org/10.18517/ijaseit.6.3.836, 2016.

Zulriskan, A. P., Koestoer, R. H., and Alwini, A. F.: Disaster mitigation of climate change effects on small islands (case of Harapan and Kelapa Islands), E3S Web Conf., 74, 12004, https://doi.org/10.1051/e3sconf/20187412004, 2018.

## Appendix B: Not considered literature

**Identified literature that could not be accessed:**

Diposaptono S., Pratikto W.A., Mano A. (2004). Flood in Jakarta - lessons learnt from the 2002 flood. In: Goda, Y., Kioda, W., Nadaoka, K. (eds.) (2004): Asian and Pacific Coasts 2003. DOI: 10.1142/9789812703040_0006.

Karyono T.H., Melyan N.H., Salsa S.Y., Fariz E. (2017). Flood Responsive Design of the Low-Income Settlements in Kampung Melayu, Jakarta, Indonesia.In: Karyono, T.H., Vale, R., Vale, B. (eds.) (2017). Sustainable Building and Built Environments to Mitigate Climate Change in the Tropics.. DOI: 10.1007/978-3-319-49601-6_12.

Karyono T.H., Bachtiar F. (2017). Adapting City for frequent floods: A case study of Jakarta, Indonesia. In: Karyono, T.H., Vale, R., Vale, B. (eds.) (2017). Sustainable Building and Built Environments to Mitigate Climate Change in the Tropics. DOI: 10.1007/978-3-319-49601-6_8.

Karyono T.H., Burhanudin D., Timothi B. (2017). Sustainable fishing settlement in Muara Angke, North Jakarta. In: Karyono, T.H., Vale, R., Vale, B. (eds.) (2017). Sustainable Building and Built Environments to Mitigate Climate Change in the Tropics. DOI:10.1007/978-3-319-49601-6_10.

Istiani, M.R. (2016). The association of traits personality and pro-social behavior among volunteers in Jakarta. DOI: 10.1166/asl.2016.6752.

**Identified literature that could not be found:**

Van Voorst, R., Handgraaf (2012). Coping with floods in a riverbank-settlement in Jakarta Indonesia The influence of material and cognitive indicators on human actor's risk behavior.

Unknown author (2018). Indonesia: Producers want protection against illegal imports.

## Appendix C: Literature categories and counts (Scopus 2000-2019)

| Code | Topic | Sub topics | Resulting Publications (2000- 2019) |
|---|---|---|---|
| 1 | **Soft factors of adaptation** | psychology | **24** |
| | | behavior | |
| | | culture | |
| | | understanding of risk | |
| | | vulnerability analysis | |
| | | framing of flood and subsidence | |
| | | willingness to pay for ecosystem services of river communities | |
| | | participation in flood control strategy planning | |

| 2 | Policy and legal analysis | institutional analysis | 12 |
| | | national policy analysis | |
| | | legal framework | |
| | | political economy of flood protection | |
| 3 | Hard adaptation | Great Garuda Project | 36 |
| | | lakes and rainwater harvesting | |
| | | polder | |
| | | dikes and flood barriers | |
| | | embankments | |
| | | river diversions | |
| 4 | Flood models & flood mapping | Precipitation models | 78 |
| | | Subsidence models | |
| | | flood loss estimation models | |
| | | urban drainage model | |
| | | flood cost analysis | |
| | | urban expansion and effects | |
| | | Sea level rise models | |
| | | community-based flood risk mapping | |
| | | shoreline retreat model | |
| 5 | Land-use (change) impact on flooding | criticality of watershed | 9 |
| | | land-use change assessment and impacts | |
| 6 | New data types | Social media | 15 |
| | | Big Data | |
| | | crowd-sourcing | |
| | | e-participation | |
| | | high-resolution data | |
| | | PetaJakarta project | |
| 7 | Watershed management and water governance | qualitative analysis of reasons for flooding | 17 |
| | | water pollution | |
| | | drinking water source analysis/model | |
| 8 | Soft and hybrid adaptation | local/community-based adaptation | 40 |
| | | firms and adaptation | |
| | | resettlement/relocation | |
| | | alternative energy sources | |
| | | disaster management | |
| | | urban adaptation planning | |
| 9 | Early Warning | GIS-based EWS | 7 |

| | | risk communication | |
|---|---|---|---|
| | | information needs during disasters | |
| 10 | **Decision support systems** | DST for location of warehouses | **5** |
| | | Disaster Information Management System | |
| | | socio-economic vulnerability index (SEVI) + MCA | |
| | | hydrological infrastructure flood vulnerability index (HIFVI) | |
| | | Integrated Assessment Framework (IAF) for subsidence | |
| 11 | **Qualitative risk descriptions** | subsidence types | **13** |
| | | flood impact | |
| x | **No link to flooding or Jakarta** | | **70** |
| | **Conference proceedings** | | **14** |

**Appendix D: Coding scheme in MaxQDA**

- Methodology/research design
- Location of flooding
- Root causes for flood risk
  - o Socio-economic causes
o Political/structural causes
  - o Environmental/physical causes
- Coping or adaptation strategy/measure
  - o Hybrid approach
  - o Non-structural/soft measures
o Structural/hard/physical measures
- Flood governance system
- Needs and/or suggestions
- Gaps and/or persisting problems

**Author contribution**

Matthias Garschagen and Mia Wannewitz designed the study. Mia Wannewitz performed the literature analysis. Mia Wannewitz and Matthias Garschagen drafted the manuscript.

**Competing interests**

The authors have no competing interests.

**Acknowledgements**

This research has received funding from the TRANSCNED project sponsored by the German Federal Ministry of Education and Research (BMBF; grant no. 01LN1710A1).

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
