# Peer review of "Mapping the adaptation solution space – Lessons from Jakarta"

_Natural Hazards and Earth System Sciences, 2020_

## Referee Comment (RC1) · Anonymous Referee #1 · 20 Nov 2020

General comments

The manuscript addresses a relevant scientific topic within the scope of NHESS. Scoping literature on the nexus of flood and Jakarta is relevant and provides a new contribution to the literature. I particularly like that the empirical analysis is embedded into a concise theoretical debate on adaptation solution spaces. The title and the abstract clearly highlight the content of the manuscript. The manuscript is well written and easy to understand. The authors provide a concise presentation of the data by appropriate graphics and summaries of the main results. The conclusion is also plausible. The appendix delivers a transparent overview of the empirical material. Besides some minor

suggestions for clarifications, my only main concern is the selection of the keywords for the literature scoping. Therefore, I recommend a minor revision to address small suggestions before getting published in the journal.

I encourage the authors to address the following recommendations:

Specific comments

Abstract: Line 14 states that literature related to "retreat from exposed areas" are barely existing. I am wondering whether you considered literature on the contested relocation policy of the Jakarta policy. Lines 20-21: I suggest to emphasize that Jakarta is not only heavily researched but also one of the most vulnerable megacities worldwide. I think this can strengthen the justification of taking Jakarta as an example. Introduction: Line 25: The title only mentions coastal cities. Why do you speak here about "coastal and delta cities"? Line 37: You state that problem of flooding and its driver are well researched. However, there is only one reference. I suggest adding more references to underpin your argument. Line 69-71: Please add a brief overview of your methodological approach.

Methods and data Line 87: I am wondering why you only used the keywords "flood" and "Jakarta". As your focus is on adaptation and coastal hazards, I assume that you turned a blind eye to papers related to "adaptation" and "coastal hazards". Line 89: I suggest to discuss the limitation of your research that particularly papers in Indonesian have not been included in the analysis. There seems to be a considerable amount of literature in Indonesian language as many Indonesian research do not publish in English. Line 89: Please explain in more detail how you derived these thematic categories and not other categories. Figure 2: Please delete the title within the graphic. The subtitle also mentions the title (the same applies to all other figures). Please add the legend of the Y-Axis "Number of publications"

Appendix: Line 562: I suggest to delete the French reference as you mentioned that only English literature had been included in the analysis.

[Figure]

Technical corrections: Line 146: run-off Line 403: Please explain Jabodetabek for readers you are not aware of the abbreviation of Jakarta's metropolitan area.

---

## Referee Comment (RC2) · Anonymous Referee #2 · 26 Jan 2021

The paper presents a summary of scientific papers recognised in the Scopus database using "Jakarta" and "flood" as keywords, with the aim of mapping the "solution space" for mitigation or adaptation strategies to cope with the long-standing and thorny problem of flooding for the city of Jakarta.

I have major concerns with the paper for the reasons reported below.

1) SUITABILITY TO NHESS JOURNAL

Consider that NHESS is a journal for high-quality studies and original research on natural hazards and their consequences. The design, implementation, and critical evaluation of mitigation and adaptation strategies are included, but the present paper only

reports a confuse collections of mitigation and adaptation strategies found in bibliographic items about floods in Jakarta. I stress that this piece is not a critical evaluation of mitigation and adaptation strategies as it lacks the rigour and the in-depth analyses that are necessary ingredients of a "critical evaluation". It is rather a jumble of contrasting opinions, which share the goal of criticizing any possible solution to flooding in Jakarta. The final solution supported by the Authors is nothing more than a praise of as saving as vague "hybrid adaptation approaches".

2) GENERAL SCOPE

In the title, "lessons from Jakarta for other coastal cities" is inappropriate. I suggest something as "Mapping the solution space for adaptation and protection from flood in Jakarta". While it is obvious that a good work in a specific context can be of inspiration (and provide lessons) for other similar situations, this aspect must not be referenced in the title, as the present paper is not intended, nor is structured, to draw general conclusions to be applied to other coastal cities. It only assesses (with significant limitations) the specific case of Jakarta, and I do not see much broader implications.

3) METHOD OF ANALYSIS

The paper analyses the number of papers in Scopus dealing with different approaches and solutions. I don't feel that the number of scientific papers is a good criterion to judge the attention given to different approaches and solutions, nor the number of papers can actually determine adaptation policies. The number of papers on specific aspects could simply indicate that some issues are multifaceted and more complex than other, thus deserving greater effort and more studies. I believe that it is more complex to assess flood hazard with the due effectiveness, accuracy and reliability, than for example assessing the exposure of people and assets. It is more difficult, and more important, to assess the real mechanisms of flood hazard correctly, then considering the uncertain future scenarios associated to climate and land use change scenarios. In this view, it seems perfectly natural to see (let's say) ten papers dealing with hydrology

and the physics of flooding, and two paper on future hypothetical scenarios.

Furthermore, I believe that literature reviews (as the present paper actually is) should look at the scientific literature realistically. It is necessary to consider the biases that unavoidably affect the scientific production before drawing conclusions. For example, it is well-known that scientists are led to increase their scientific production enormously, with an increasing number of articles and with an inevitable reduction in research quality. The plot of Figure 2, which show an increase of papers focusing on Jakarta and floods, should be compared to the trend of research papers in the same field (e.g., concerning only "flood").

Finally, according to the two previous points, I stress that judgements based on the number of papers should be avoided (or, at least, significantly limited) in the present paper, and the attention should always be brought back to the contents of scientific papers. In other words, a single paper reporting a comprehensive analysis is more important than 20 paper written to enlarge the publication record of authors eager for career advancement.

4) MISSING BACKGROUND INFORMATION

For a reader that does not know much of flood hazard in Jakarta, it is difficult to forge a proper idea about the different countermeasures to flood hazard/risk listed in the paper. A paragraph should be added that summarize the main source of risk in Jakarta (e.g., coastal and/or river flooding), the areas interested by each different flooding mechanisms, the mean flow depth that is expected. This is a fundamental aspect because, for example, "soft" measures are almost useless in the case of frequent flooding with water depth of more than 1 m (either you leave the area, or you keep water away, no half measures); completely different is the case of nuisance flooding.

5) OUTCOMES OF THE STUDY

Basically, previous literature studies are divided in two families: engineering pieces

dealing with physical aspects and infrastructural solutions from a "protection from flooding" perspective, that are associated to technocratic solutions, and (generally qualitative) pieces dealing with soft adaptation and associated to bottom-up, more sustainable and comprehensive solutions. The first approach is criticized as outdated and possibly ineffective. The second approach is criticized as lacking concrete recommendations on how to achieve better goals.

I feel that the paper elaborates on a great misunderstanding. Structural interventions unavoidably entail negative impacts, but here the criticism of design choices because, as it seems, they are associated to the interests of the wealthiest classes, is confused with the criticism of technical solutions.

In summary, on one hand the paper turns out to be a critique of the Great Garuda Project (a major structural intervention that is to be built). I do not claim that the Great Garuda Project is a right choice or not, the problem here is that the motivation against the Great Garuda Project are not clearly reported nor analysed in the paper! The alternatives to the Great Garuda Project, and to the classic engineering approach of "protection from flood", are extremely vague, unsubstantiated, not analyzed in depth and, indeed, of dubious utility considering the extent of the flood risk. Indeed, the analysis neglects a fundamental aspect: soft measures are almost useless against hard flooding.

Much of the conclusions reported in the paper are not supported, nor they are the logical conclusion of the given premises. For example (l. 451-453) "the pursuit of such infrastructural measures despite their questionable effectiveness and major critique shows that the city government sticks to its traditional protection approach". Are there effective alternatives to infrastructural measures? This issue is not clearly addressed in the paper, so the conclusion that "the city government sticks to its traditional protection approach" is not the logical consequence. If an "outdated" structural measure is the only effective solution to a present problem, even a government devoted to the future would be obliged to choose this one.

MINOR COMMENTS

-l. 29: Tellmann et al. (2020) and Wolff et al. (2020) are missing in the bibliography.

-l. 130: Figure 4, not 3.

-l. 324: "While there hence exists. . ." is an awkward construction.

-l. 419: please introduce DRR acronym.

-l. 773: the title of the paper is repeated two times.

-An analysis of a coastal area affected by land subsidence, flooding and population dynamics, is reported in https://doi.org/10.1016/j.scitotenv.2018.09.121

-Two examples of adaptation measures supported by technical studies are https://doi.org/10.1016/j.ejrh.2020.100702 and https://doi.org/10.3390/w12061609

---

## Author Comment (AC1) · 16 Mar 2021

We would like to thank Referee 1 for the constructive comments, which we consider to be very helpful for improving the manuscript. Overall, we agree to the points raised and the suggestions provided, which have been taken up in the revision of the paper. The below report provides a detailed account of our responses to the individual comments. Main changes to the manuscript include an extension of the search term combination to complement the conducted literature search as well as a revision of the methodology chapter that will then also touch upon epistemological underpinnings to explain the development of the literature categories.

**1. General comments**

RC: The manuscript addresses a relevant scientific topic within the scope of NHESS. Scoping literature on the nexus of flood and Jakarta is relevant and provides a new contribution to the literature. I particularly like that the empirical analysis is embedded into a concise theoretical debate on adaptation solution spaces. The title and the abstract clearly highlight the content of the manuscript. The manuscript is well written and easy to understand. The authors provide a concise presentation of the data by appropriate graphics and summaries of the main results. The conclusion is also plausible. The appendix delivers a transparent overview of the empirical material. Besides some minor suggestions for clarifications, my only main concern is the selection of the keywords for the literature scoping. Therefore, I recommend a minor revision to address small suggestions before getting published in the journal.

Answer: We would like to thank Referee 1 for the review and the constructive suggestions. The keyword search had been kept very open in the original manuscript in order to cast the net as widely as possible. However, we agree that this issue deserves more explicit explanation and triangulation. We therefore superimpose a second keyword search in the revised version of the paper and discuss the outcomes of this search in detail, taking on board the few additional papers that the second search has yielded.

**2. Abstract**

RC: Line 14 states that literature related to "retreat from exposed areas" are barely existing. I am wondering whether you considered literature on the contested relocation policy of the Jakarta policy.

Answer: Thank you for this comment. Indeed, the structurally reviewed literature covered retreat from exposed areas through Jakarta's relocation policies, which is strongly criticized by the publications dealing with this particular issue. However, what did barely come up in the reviewed literature are successful and socially just retreat and/or relocation options. We have clarified this aspect in the revised version of the manuscript.

RC: Lines 20-21: I suggest to emphasize that Jakarta is not only heavily researched but also one of the most vulnerable megacities worldwide. I think this can strengthen the justification of taking Jakarta as an example.

Answer: Thank you for this helpful suggestion. The revised version highlights that argument more strongly in the abstract and introduction, drawing on comparative global studies.

3. Introduction

RC: Line 25: The title only mentions coastal cities. Why do you speak here about "coastal and delta cities"?

Answer: Thank you. The reference to delta cities has been deleted as the paper does not explicitly discuss the adaptation needs and options of delta cities which to some extent vary from other coastal cities.

RC: Line 37: You state that problem of flooding and its driver are well researched. However, there is only one reference. I suggest adding more references to underpin your argument.

Answer: Thank for this valid observation. There is indeed ample literature on flooding and flood drivers in Jakarta. Additional key references have been added.

RC: Line 69-71: Please add a brief overview of your methodological approach.

Answer: A brief overview over the methodological approach has been added to the introduction. In addition, the revision of the methods chapter includes a clearer description of the development of the literature categories, including the conceptual and epistemological underpinnings.

4. Methods and data

RC: Line 87: I am wondering why you only used the keywords "flood" and "Jakarta". As your focus is on adaptation and coastal hazards, I assume that you turned a blind

eye to papers related to "adaptation" and "coastal hazards".

Answer: Thank you for this very valid comment. We chose a very broad search term combination "flood*" AND "Jakarta" under the assumption that papers elaborating on "coastal hazards" and "adaptation" would mention these terms in their title or abstracts in any case. However, to crosscheck, we conducted an additional search with the search term combination Jakarta AND (flood* OR adaptation OR coastal hazard). There was considerable overlap with our initial search (238 paper overlapping) but 22 new highly relevant and 12 relevant papers were identified and will be added to the review.

RC: Line 89: I suggest to discuss the limitation of your research that particularly papers in Indonesian have not been included in the analysis. There seems to be a considerable amount of literature in Indonesian language as many Indonesian research do not publish in English.

Answer: Thank you. While the paper briefly mentions that only English literature was included, we agree that the reflection on the limitations of this approach was underdeveloped in the original manuscript. The revised version features a stronger and more explicit discussion.

RC: Line 89: Please explain in more detail how you derived these thematic categories and not other categories.

Answer: Thank you for this important observation. The revised methods section features a clearer explanation of our approach, particularly with respect to the important question of how we arrived at our thematic categories for the analysis.

5. Main

RC: Figure 2: Please delete the title within the graphic. The subtitle also mentions the title (the same applies to all other figures). Please add the legend of the Y-Axis "Number of publications".

Answer: Thank you. All figures are edited following this advice.

RC: Line 146: run-off

Answer: The spelling mistake has been corrected.

RC:Line 403: Please explain Jabodetabek for readers you are not aware of the abbreviation of Jakarta's metropolitan area.

Answer: Thank you. An explanation of the term Jabodetabek has been added.

6. Appendix

RC: Line 562: I suggest to delete the French reference as you mentioned that only English literature had been included in the analysis.

Answer: Thank you for this advice. French references have been deleted in the revised manuscript.

---

## Author Response (AR1)

| | Comments | Submitted Answer | Changes in the manuscript |
|---|---|---|---|
| | | **Review 1** | |
| **General comments** | The manuscript addresses a relevant scientific topic within the scope of NHESS. Scoping literature on the nexus of flood and Jakarta is relevant and provides a new contribution to the literature. I particularly like that the empirical analysis is embedded into a concise theoretical debate on adaptation solution spaces. The title and the abstract clearly highlight the content of the manuscript. The manuscript is well written and easy to understand. The authors provide a concise presentation of the data by appropriate graphics and summaries of the main results. The conclusion is also plausible. The appendix delivers a transparent overview of the empirical material. Besides some minor suggestions for clarifications, my only main concern is the selection of the keywords for the literature scoping. Therefore, I recommend a minor revision to address small suggestions before getting published in the journal. | We would like to thank Referee 1 for the review and the constructive suggestions. The keyword search had been kept very open in the original manuscript in order to cast the net as widely as possible. However, we agree that this issue deserves more explicit explanation and triangulation. We therefore superimpose a second keyword search in the revised version of the paper and discuss the outcomes of this search in detail, taking on board the few additional papers that the second search has yielded. | The comments of Reviewer 1 were very valuable and duly considered in the revision of the manuscript. The reviewers main concern regarding the search terms was addressed by conducting a second key word search, which led to the identification of a few additional papers that were added to the analysis. |
| **Abstract** | Line 14 states that literature related to "retreat from exposed areas" are barely existing. I am wondering whether you considered literature on the contested relocation policy of the Jakarta policy. | Thank you for this comment. Indeed, the structurally reviewed literature covered retreat from exposed areas through Jakarta's relocation policies, which is strongly criticized by the publications dealing with this particular issue. However, what did barely come up in the reviewed literature are successful and socially just retreat and/or relocation options. We have clarified this aspect in the revised version of the manuscript. | To clarify this point, we edited the respective statement in the abstract, which now reads that there is only little literature on socially just relocation. |
| | Lines 20-21: I suggest to emphasize that Jakarta is not only heavily researched but also one of the most vulnerable megacities worldwide. I think this can strengthen the justification of taking Jakarta as an example. | Thank you for this helpful suggestion. The revised version highlights that argument more strongly in the abstract and introduction, drawing on comparative global studies. | To address this aspect, we edited the abstract, introduction and conclusion by highlighting Jakarta's comparably high exposure and vulnerability to flooding. |
| | Line 25: The title only mentions coastal cities. Why do you speak here about "coastal and delta cities"? | Thank you. The reference to delta cities has been deleted as the paper does not explicitly discuss the adaptation needs and options of delta cities which to some extent vary from other coastal cities. | We deleted "delta" throughout the document to be consistent, as suggested by the reviewer. |
| | Line 37: You state that problem of flooding and its driver are well researched. However, there is only one reference. I suggest adding more references to underpin your argument. | Thank for this valid observation. There is indeed ample literature on flooding and flood drivers in Jakarta. Additional key references have been added. | Following the reviewer's comment, we added literature on flooding and flood drivers, showcasing that Jakarta is well researched in this respect. |
| | Line 69-71: Please add a brief overview of your methodological approach. | A brief overview over the methodological approach has been added to the introduction. In addition, the revision of the methods chapter includes a clearer description of the development of the literature categories, including the conceptual and epistemological underpinnings. | Addressing this comment, we added a brief sentence on the methodology in the abstract as well as in the introduction. In addition, we revised the methodology chapter to make our methodological approach clearer and more transparent. |
| **Methods and data** | Line 87: I am wondering why you only used the keywords "flood" and "Jakarta". As your focus is on adaptation and coastal hazards, I assume that you turned a blind eye to papers related to "adaptation" and "coastal hazards". | Thank you for this very valid comment. We chose a very broad search term combination "flood*" AND "Jakarta" under the assumption that papers elaborating on "coastal hazards" and "adaptation" would mention these terms in their title or abstracts in any case. However, to crosscheck, we conducted an additional search with the search term combination Jakarta AND (flood* OR adaptation OR coastal hazard). There was considerable overlap with our initial search (238 paper overlapping) but 22 new highly relevant and 12 relevant papers were identified and will be added to the review. | We followed the valuable comment of the reviewer and edited out search terms. The new search term combination Jakarta AND ("flood*" OR "coastal hazard*" OR "adaptation") resulted in 311 publications (2000-2019, excl. non-relevant subject areas). We integrated insights from an additional 22 highly relevant and relevant publications into the analysis, following the same methodological approach as described in the methodology chapter. |

| | | | |
|---|---|---|---|
| | Line 89: I suggest to discuss the limitation of your research that particularly papers in Indonesian have not been included in the analysis. There seems to be a considerable amount of literature in Indonesian language as many Indonesian research do not publish in English. | Thank you. While the paper briefly mentions that only English literature was included, we agree that the reflection on the limitations of this approach was underdeveloped in the original manuscript. The revised version features a stronger and more explicit discussion. | To address the reviewer's concern, we raise and discuss this limitation more explicitly it in the methodology chapter. |
| | Line 89: Please explain in more detail how you derived these thematic categories and not other categories. | Thank you for this important observation. The revised methods section features a clearer explanation of our approach, particularly with respect to the important question of how we arrived at our thematic categories for the analysis. | We considered this comment by heavily revising our methodlgy chapter. We added a table, which gives an overview of the literature categories. In addition, we explain the research process and the development of the literature categories in a more detailed manner to increase transparency. |
| **Main** | Figure 2: Please delete the title within the graphic. The subtitle also mentions the title (the same applies to all other figures). Please add the legend of the Y-Axis "Number of publications" | The figures were edited accordingly. | |
| | Line 146: run-off
Line 403: Please explain Jabodetabek for readers you are not aware of the abbreviation of Jakarta's metropolitan area. | The spelling mistake was corrected and the Jabodetabek is now explained in a footnote. | All changed accordingly. |

| Appendix | Line 562: I suggest to delete the French reference as you mentioned that only English literature had been included in the analysis. | French references are deleted. | |
|---|---|---|---|
| | | **Review 2** | |
| **Suitability of journal/ general comments** | Consider that NHESS is a journal for high-quality studies and original research on natural hazards and their consequences. The design, implementation, and critical evaluation of mitigation and adaptation strategies are included, but the present paper only reports a confuse collections of mitigation and adaptation strategies found in bibliographic items about floods in Jakarta. | We take note of the reviewer's concern and have revised the manuscript to more clearly explain our approach and its contribution to the generation of knowledge in the field of natural hazards and risk reduction. Overall, and in line with the feedback received from referee #1, we are convinced that the structured review and assessment of the peer-reviewed academic literature can make a significant contribution to analyzing the state of an academic debate, in this case around the solutions discussed for Jakarta's flood problem. In this sense, we consider the paper to be very much in line with the scope and aim of NHESS to "serve a wide and diverse community of research scientists, practitioners, and decision makers concerned with […] the design and implementation of mitigation and adaptation strategies, including economical, societal, and educational aspects". Our approach to review, assess and synthesize the scientific literature in a structured manner and against a set of analytical questions is a standard approach in academia. It is a pity that the reviewer is left with the impression that the paper primarily "reports a confuse collection of mitigation and adaptation strategies found in bibliographic items". At the same time, we take this conception seriously and have therefore heavily revised the paper to explain and discuss much more clearly the objectives, underlying concepts and steps of our approach as well as its shortcomings and limits, e.g. with respect to potential gaps between was is reported in the English scientific literature and what might be discussed in policy and practice outside of this body of data. | To address the reviewer's comment, we heavily revised the entire manuscript. In particular, the methodology chapter and the structure of the results were re-worked in order to improve transparency of the research process as well as to provide a more consise overview of our findings. With regard to the evaluation of identified adaptation options, we now clearly separate between evaluations as found in the literature (results chapter) and own evaluation (discussion chapter). In addition, a new conceptual chapter explains our perspective on risk and adaptation solutions, providing a sound foundation for our analysis. Furthermore, the limitations of the structured literature were stated more clearly. |
| | I stress that this piece is not a critical evaluation of mitigation and adaptation strategies as it lacks the rigour and the in-depth analyses that are necessary ingredients of a "critical evaluation". | We take note of this comment. However, the paper does not claim to provide an own critical evaluation of all mitigation and adaptation strategies discussed in the scientific literature. The paper aims to respond to the questions clearly laid out in lines 66-68. At the same time, we take the conception of the reviewer seriously and have heavily revised the manuscript to avoid the impression that an own evolution of adaptation options – and especially own opinions – is our primary objective. Revisions pertain particularly to the introduction and conclusion section as well as to language edits throughout the entire paper. | Considering the comment, we revised the result and discussion chapter. A clear separation of how different adaptation options are portrayed and evaluated in the literature (results chapter) and how we would evaluate them in relation to the adaptation solution space they span (discussion chapter) is supposed to provide a clear structure and facilitate reader guidance. In addition, we changed missleading wording throughout the manuscript. |
| | It is rather a jumble of contrasting opinions, which share the goal of criticizing any possible solution to flooding in Jakarta. | Remove critical wording and comments on physical/infrastructural solutions. | We followed the recommendation of the reviewer by editing critical wording of infrastructural adaptation measures throughout the manuscript. |

| | | | |
|---|---|---|---|
| | The final solution supported by the Authors is nothing more than a praise of as saving as vague "hybrid adaptation approaches". | We take note of the comment but it is not quite clear what the reviewer is aiming at. The conclusion in fact summarizes how the assessed literature on Jakarta treats hybrid solutions – and it observes that the gap of that treatment is noteworthy when juxtaposed against the co-benefits of such hybrid solutions, as reported in the literature more generally. We have difficulty to see how such a conclusion is wrong or problematic. However, we have expanded the review of those articles discussing hybrid solutions for Jakarta and the conceptual framing around hybrid solutions in the newly added conceptual section. Against this background, we have also sharpened the conclusion regarding hybrid solutions. | In due consideration of the reviewer's critique, we have expanded our manuscript with a conceptual chapter that not only clarifies our conception of risk but also of different adaptation options, including hybrid adaptation and what this appraoch entails. What is more, we strengthened our notion of hybrid adaptation options as described in the literature in the results section and picked up on it again in our discussion. |
| **General scope** | In the title, "lessons from Jakarta for other coastal cities" is inappropriate. I suggest something as "Mapping the solution space for adaptation and protection from flood in Jakarta".
While it is obvious that a good work in a specific context can be of inspiration (and provide lessons) for other similar situations, this aspect must not be referenced in the title, as the present paper is not intended, nor is structured, to draw general conclusions to be applied to other coastal cities.
It only assesses (with significant limitations) the specific case of Jakarta, and I do not see much broader implications. | Thank you for this useful observation. The lessons for other coastal cities is meant not in the sense that the situation of or academic engagement with flood adaptation in Jakarta can easily be transferred to other coastal cities. Rather, the identified patterns and gaps are meant to stimulate similar assessments of the current debate in other contexts. Judging from the literature on other coastal cities, at least in Southeast Asia, there are indications that some of the patterns found here are also true for other coastal cities in the region. Triggering a more detailed look is the main objective behind the "lessons" argument. The paper has been revised to clarify this point, particularly in the introduction section, the newly added conceptual section and the conclusions. In addition, the title has been adjusted accordingly. | To address the reviewer's points, the manusicript now highlights more clearly how the case of Jakarta represents a valuable example for other coastal cities in Southeast Asia which face similar issues. In the conclusion, we highlighted more dedicatedly that our findings cannot be translated directly to other coastal cities but how this study may inspire similar analyses in highly at-risk coastal cities with the broader objective to widen their respective adaptation solution space. Strengthening the relevance of this case study, we decided to stick to the original title. |
| **Method** | I don't feel that the number of scientific papers is a good criterion to judge the attention given to different approaches and solutions, nor the number of papers can actually determine adaptation policies. | The reviewer is right, of course, in that the number of scientific papers does not necessarily translate into the level of attention given to a certain topic in general or the policies in that field – this is obvious. The paper therefore does not claim to assess this link. The paper works towards answering the research questions laid out in lines 66-68! These questions are concerned with how different adaptation options are perceived and framed in the academic debate. Here we believe that publication intensity on certain types of measures is one indicator – amongst others. We have revised the paper to explain this approach more clearly, hoping to avoid misconceptions. In addition, however, we added a discussion on whether and how scientific problem framing can in fact contribute to the framing of problems and solutions outside of academic realms. There is a long-established scientific literature on this inquiry, see for instance the discussion of the "dominant view" of risk reduction in the second half of the past century (Hewitt 1983). | Taking the comment in due consideration, we added a disclaimer in the methodology chapter that the number of publications alone does not determine adaptation policies. At the same time we highlighted that it can serve as an indicator for how the solution space is shaped. Explanations were added accordingly in the methods and conclusion section. |

| | | | |
|---|---|---|---|
| | The number of papers on specific aspects could simply indicate that some issues are multifaceted and more complex than other, thus deserving greater effort and more studies. I believe that it is more complex to assess flood hazard with the due effectiveness, accuracy and reliability, than for example assessing the exposure of people and assets. It is more difficult, and more important, to assess the real mechanisms of flood hazard correctly, then considering the uncertain future scenarios associated to climate and land use change scenarios. | Thank you for this though-provoking comment. Complex problems might of course trigger more publications. However, we are not convinced that flood hydrology per se is a more complex scientific problem than, say, the assessment of future trends in socio-economic vulnerability in highly dynamic contexts such as Jakarta. One could also argue the other way around: There are established data sets and methodological approaches to model and assess a city's flood hydrology, which might suggest that in fact less publications are needed to tackle this topic – very much in contrast to more open and emerging fields. The point is: The number of publications is an indicator that has to be interpreted with much care. We have strengthened the manuscript to discuss these questions more thoroughly. | see above |
| | In this view, it seems perfectly natural to see (let's say) ten papers dealing with hydrology and the physics of flooding, and two paper on future hypothetical scenarios. | see above | see above |
| | Furthermore, I believe that literature reviews (as the present paper actually is) should look at the scientific literature realistically. It is necessary to consider the biases that unavoidably affect the scientific production before drawing conclusions. For example, it is well-known that scientists are led to increase their scientific production enormously,with an increasing number of articles and with an inevitable reduction in research quality. | The response to this comment builds on our response to the previous comment. The amount of publications on a certain topic – in this case adaptation measures – can depend on many factors. These factors do not only include the aspects mentioned by the reviewer but also other issues such as the numbers of post-graduate students in different disciplines, different publication styles in different disciplines, the availability of data sets etc. We have added a dedicated paragraph discussing these factors and their relation to our results in section 4. The authors are not aware of large-n empirical studies in support of the sweeping statement made by the reviewer that the push towards increased academic output "inevitably" reduces research quality. | A paragraph stating the limitations of the approach has been added in the beginning of the results chapter. |
| | The plot of Figure 2, which show an increase of papers focusing on Jakarta and floods, should be compared to the trend of research papers in the same field (e.g., concerning only "flood"). | Thank you for this helpful suggestion. We will revise the graph accordingly. | We followed the reviewer and conducted an additional search in Scopus ("flood* OR "coastal hazard*" OR "adaptation"). The publication trend was added to the figure as suggested and clearly depicts the difference between the global trend on flood risk research and the trend of respective research focusing on Jakarta. |
| | Finally, according to the two previous points, I stress that judgements based on the number of papers should be avoided (or, at least, significantly limited) in the present paper, and the attention should always be brought back to the contents of scientific papers. | Thank you for this observation. The language has been revised throughout the manuscript and dedicated information added in cases similar to those one outlined by the reviewer. | Considering this point, we clarified in the methodology chapter that we did not only consider numbers of publications but also their content. In fact, we explicitly focused on publications touching on less well covered topics to compensate less popular fields to a certain extent. |
| | In other words, a single paper reporting a comprehensive analysis is more important than 20 paper written to enlarge the publication record of authors eager for career advancement. | see above | see above |
| **Background information** | For a reader that does not know much of flood hazard in Jakarta, it is difficult to forge a proper idea about the different countermeasures to flood hazard/risk listed in the paper. | Thank you for this constructive suggestion. The revised version of the paper now features a chapter introducing the reader to the hazard context of Jakarta. It provides a brief overview of drivers and causes of flood risk in the city and it shortly describes consequences and effects of | Duly following the reviewer's comment, we added a brief overview Jakarta's flood context to the manuscript. It presents causes of flooding, impacts and recent flood risk management policies to introduce the reader to the specific situation of Jakarta. |

| | | | |
|---|---|---|---|
| | A paragraph should be added that summarize the main source of risk in Jakarta (e.g., coastal and/or river flooding), the areas interested by each different flooding mechanisms, the mean flow depth that is expected. | some of the most recent flood events. | |
| | This is a fundamental aspect because, for example, "soft" measures are almost useless in the case of frequent flooding with water depth of more than 1 m (either you leave the area, or you keep water away, no half measures); completely different is the case of nuisance flooding. | This is a relevant comment if you consider adaptation measures to only comprise those measures that directly take effect on the flood hazard. However, adaptation measures go far beyond this realm, covering, for example, things like knowledge provision or the strengthening of social safety nets. Here, soft measures bear relevance also in higher flood scenarios. In order to clarify these points conceptually, we have added a conceptual section at the beginning of the paper. Besides detailing on our understanding of risk and other key concepts, it covers what is understood as "soft" and "hard" adaptation measures. | We addressed this comment by detailing conceptual underpinnings of how we understand risk and what different adaptation options, i.e. hard, soft and hybrid, are and entail in a new conceptual chapter. It explains our adopted perspective on risk and portrays hard, soft and hybrid adaptation options as part of a comprehensive response to flooding. |
| Outcomes | I feel that the paper elaborates on a great misunderstanding. Structural interventions unavoidably entail negative impacts, but here the criticism of design choices because, as it seems, they are associated to the interests of the wealthiest classes, is confused with the criticism of technical solutions. | We thank the reviewer for this thought-provoking comment. From our point of view, the overall success of a measure can be evaluated from different perspectives – and disentangling negative impacts can be quite complicated in a complex political economy such as the one of Jakarta. Technical solutions can be effective in avoiding flooding and hence be evaluated as successful. However, as the case of Jakarta shows, the implementation of such technical measures are often accompanied by the eviction of highly vulnerable groups. In the end, this can increase vulnerability to flooding of certain groups instead of reducing it. In other words, the separation between critique on "design choices" of technical solutions and critique on technical solutions overall might not be as simple as suggested by the reviewer. However, these considerations show that the issue indeed needs a more explicit discussion in the paper. We have revised the manuscript to add this discussion accordingly. | In consideration of the reviewer's comment, we have edited the result section on hard adaptation measures as well as the discussion and conclusion chapter. Language was revised to differentiate more clearly between critique on the effectiveness of daptation measures and potentially negative impacts due to design choices. The issue is also discussed more dedicately in the results and discussion section. |
| | In summary, on one hand the paper turns out to be a critique of the Great Garuda Project (a major structural intervention that is to be built). I do not claim that the Great Garuda Project is a right choice or not, the problem here is that the motivation against the Great Garuda Project are not clearly reported nor analysed in the paper! | This is an interesting comment. We would like to clarify that our study is not meant to be a critique to the Great Garuda Project. We assess how different adaptation measures are being reflected in the academic literature. With the Great Garuda Project being the single largest flood risk reduction measure in Jakarta it is not surprising that it receives significant attention also in the scientific literature. What is more, this being a highly contested measures it is further not surprising that some of the literature is quite critical of it. This is mirrored in our review and assessment of the literature. However, we revised the manuscript to change any sections that could be interpreted as a critique of the Great Garuda Project driven out of a personal motivation. | To avoid the misconception that the manuscript is first and foremeost critizising hard adaptation solutions such as the Great Garuda Project, we have revised misleading language throughout the document. Critique raised in the analysed literature is now clearly separated from own evaluations to avoid any potential misinterpetation. |

| | | | |
|---|---|---|---|
| | The alternatives to the Great Garuda Project, and to the classic engineering approach of "protection from flood", are extremely vague, unsubstantiated, not analyzed in depth and, indeed, of dubious utility considering the extent of the flood risk. Indeed, the analysis neglects a fundamental aspect: soft measures are almost useless against hard flooding. | Thank you for this interesting comment. It is true that in the literature on flood risk reduction in Jakarta, one finds a lot of critique on the Great Garuda Project but less debate on what could be viable and effective alternatives. We revised the manuscript to strengthen this aspect, particularly in the discussion section. The comment on alleged uselessness of soft measures is tackled in one of our above responses. | We addressed this comment by revising the discussion in a way that it now more clearly points to the gaps identified in the course of the analysis. One of them is the lack of comparative studies of different adaptation options and the lack of research on hybrid adaptation options as potential alertntives to solely hard adaptation measures. These aspects were flagged more explicitly in the discussion section. |
| | Much of the conclusions reported in the paper are not supported, nor they are the logical conclusion of the given premises. For example (l. 451-453) "the pursuit of suchninfrastructural measures despite their questionable effectiveness and major critiquenshows that the city government sticks to its traditional protection approach". Are thereneffective alternatives to infrastructural measures? This issue is not clearly addressed in the paper, so the conclusion that "the city government sticks to its traditional protection approach" is not the logical consequence. If an "outdated" structural measure is the only effective solution to a present problem, even a government devoted to the future would be obliged to choose this one. | Thank you for this comment. We have added more nuance to the conclusions, especially with regards to the framing and evaluation of infrastructure measures, as discussed in the literature. We have also made the line of sight to the underlying assessment in the paper more visible. In addition, we revised the conclusion to more carefully differentiate between evaluations put forward in the assessed literature and own judgements. One of the key critiques raised in the literature is that infrastructure measures are not fully effective, whilst generating substantial externalities (social and ecological). At the same time, the literature does not discuss potential alternatives in great detail. We strengthened the discussion of this gap in section 4 and the conclusions. | We have added more nuance to the conclusions, especially with regards to the framing and evaluation of infrastructure measures, as discussed in the literature. We have also made the line of sight to the underlying assessment in the paper more visible. In addition, we revised the discussion to more carefully differentiate between evaluations put forward in the assessed literature and own observations. Furhtermore, we strengthened the discussion in that we emphasized the lack of researching potential adaptation altrnatives. |
| **Other** | l. 29: Tellmann et al. (2020) and Wolff et al. (2020) are missing in the bibliography.
l. 130: Figure 4, not 3.
l. 324: "While there hence exists. . ." is an awkward construction.
l. 419: please introduce DRR acronym.
l. 773: the title of the paper is repeated two times. | all considered and changed | All changed accordingly. |
| | An analysis of a coastal area affected by land subsidence, flooding and population dynamics, is reported in
https://doi.org/10.1016/j.scitotenv.2018.09.121

Two examples of adaptation measures supported by technical studies are
https://doi.org/10.1016/j.ejrh.2020.100702 and
https://doi.org/10.3390/w12061609 | The suggested references are appreciated and will be carefully considered for inclusion in the manuscript. | We duly considered the suggested references and added those evaluated as relevant to the conceptual chapter. |

---

## Author Response (AR2)

| | **Referee comments and authors' answers** | |
|---|---|---|
| | **Editor** | |
| | Dear authors,
at this stage, only minor issues remain to be fixed in your manuscript. I suggest checking carefully the suggestions provided by the reviewers. In addition, a quick comment from my side: in order to make the article more attractive for our readers, I suggest an enrichment with a map of the study area and 2-3 pictures. | Thank you very much for your kind and helpful comments. We have carefully considered your and all adiitinalreview comments and improved the paper accodngly. To your comment, we also included a map showing the main adaptation infrastructure currently under discussion for Jakarta. |
| | **Referee #1 (Report 2)** | **Answer by the authors** |
| General | The authors have carefully edited all comments and errors. The manuscript has improved significantly. Before the manuscript is accepted for publication, I recommend that a few small comments be considered, which are listed below. | Thank you very much for your positive evaluation and for taking the time to carefully reviewing the document again. We have duly considered your helpful comments in order to further improve the manuscript. |
| Line 55 | Please futher elaborate how „scientific inquiry contributes to the evolution and shape of the solution space". I think it helps to justify much better your methodological approach. | We highly appreciate your comment and have added why we belief that assessing the solution space as we do it in our analysis is a worthwhile and valuable exercise. |
| Lines 75-77 | Please provide some examples of other coastal cities at risk | Thank you very much for your comment. We have added a few examples in line 53f., as we felt it fit well in that context. |
| Line 236 | 311 papers are mentioned. In Figure 1, however, there are 340 papers stated. Which number is correct? | Thank you very much for this hint. We corrected the number in the text, which changed due to the extension of the search. |
| Lines 294-296 | I assume you mean Table 2. The percentage in the text does not match the table. Table 2 says that 48.6% of the first authors are linked to Indonesian institutions. The same is true for the number of Dutch institutions, and the ranking is different from the table. Thus, there are more German first authors than Singaporeans or US-Americans. | Thank you very much for this comment. We have carefully re-visited the numbers and corrected them accordingly. |
| | Please list all publications considered in your analysis in the reference list. Publications that have been included in the literature analysis should be marked e.g. with an asterisk or another symbol. | We appreaciate your comment and have added the full list of publications that resulted from the search in the annex. Not all resulting studies were cited in the article, which is why we refrained from simply adding them to the reference list. |
| | **Referee # 2 (Report 1)** | **Answer by the authors** |
| General | I recognize that the Authors did a good job in improving the manuscript. Now the analysis is more focused on the solution space dealt with by previous literature studies, the judgement is more balanced, and the lacking aspects of the different approaches are clearly identified. Some minor points remain to be fixed before publication. | Thanky you very much for your positive evaluation of our edits. We will carefully consider your remarks in order to further improve our manuscript. |
| Title | I remain convinced that the present study is not aimed at drawing general conclusion for other coastal cities. The focus is undoubtedly on Jakarta. As I already said, a good work can surely provide ideas for similar situations, but the title of a paper should reflect the main aim of a study. "lessons from Jakarta for other coastal cities" is not the focus of the present work (for example, there are no comparison with other mega-cities), hence this part of the title remains misleading. Considering that the solution space includes adaptation and also protection from floods, I return to suggest "Mapping the solution space for adaptation and protection from flood in Jakarta". | We appreciate your comment and have changed the title accordingly. |
| Lines 516-521 | ("While these lessons… which calls for follow-up research) are not coherent with the paragraph content and interrupt the discussion focussed on Jakarta. I suggest moving these lines to l. 532, before "Jakarta can be a globally leading pilot..." (but starting a new, last paragraph). | Thank you very much for your comment. We have edited the paragraph accordingly to improve readybility. |
| Line 139 | a reference to Mel et al. (2020), already present in the Bibliography section, is lacking. | Thank you for this hint. We have added the reference accordingly. |